# SHAREFORMER: SHARE ATTENTION FOR EFFICIENT IMAGE RESTORATION

## ABSTRACT

Transformer-based networks are gaining popularity due to their superior ability to handle long-range information. However, they come with significant drawbacks, such as long inference time, and challenging training processes. These limitations become even more pronounced when performing high-resolution image restoration tasks. We have noticed that there is a trade-off between models' latency time and their trainability. Including a convolutional module can improve the networks' trainability but not reduce their latency. Conversely, sparsification notably reduces latency but renders networks harder to optimize. To address these issues, a novel Transformer for image restoration called ShareFormer is proposed here. ShareFormer offers optimal performance with lower latency and better trainability than other Transformer-based methods. It achieves this by facilitating the sharing of the attention maps amongst neighboring blocks in the network, thereby considerably improving the inference speed. To maintain the model's information flow integrity, residual connections are added to the "Value" of self-attention. Several lesion studies indicate that incorporating residual connections on "Value" can aggregate the shallow transformers with shared attention, introducing a local inductive bias and making the network easier to optimize without the need for additional convolution. The effectiveness, efficiency, and easy-to-train of our ShareFormer is supported by numerous experimental results. Our code and pre-trained models will be open-sourced upon publication of the paper.

## 1 INTRODUCTION

Image restoration is a classical inverse problem. It aims to reproduce high-quality images from degraded (e.g., bicubic downsampling, noise, jpeg compression, etc.) inputs. Since the inception of SRCNN (Dong et al., 2015) and DnCNN (Zhang et al., 2017), CNN-based methods have been widely used to solve image restoration problems. However, as the architectures evolve, CNN-based methods (Vedaldi & Lenc, 2015; Zhang et al., 2018a;b; 2020; Zhao et al., 2020; Zhang et al., 2021) face the problems of excessive inductive bias and small network perceptual field, which limit the performance. To solve these problems, Transformer-based methods (Liang et al., 2021; Zamir et al., 2022; Wang et al., 2022; Yawei Li et al., 2023) were proposed. Converse to high-level vision tasks, image restoration tasks involving Transformers require careful

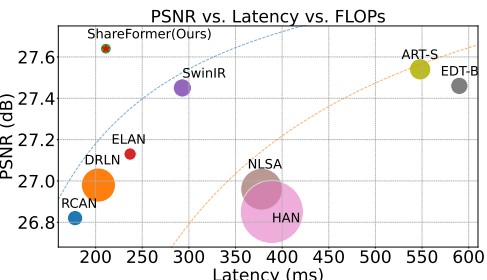

Figure 1: Comparison on ×4 image super-resolution task in terms of networks' accuracy, latency, and floating point operations (FLOPs). The area of each circle denotes the FLOPs of the network. Our method (Share-Former) achieves competitive performance while having lower latency and FLOPs.

consideration to ensure that feature resolution and window sizes of attention are sufficiently large to prevent undue compression of texture information. This requirement makes the overhead for Transformer-based networks used in image restoration tasks much higher than their usage in other tasks. Meanwhile, in recent years, performance improvement of low-level vision networks such as IPT (Chen et al., 2021a), EDT (Li et al., 2023b), and HAT (Chen et al., 2023) has increasingly relied on sophisticated training strategies such as long-scale pre-training and some other tricks (Lin et al., 2022). Thus, two questions regarding the design of Transformers are poised to arise: (a) **How to make Transformer faster?** (b) **How to make Transformer's optimization easier?**

| Methods | Self Attention (ms) | Feed Forward (ms) | Convolutions (ms) | Total (ms) | Trainability($\downarrow$) |
|---|---|---|---|---|---|
| SwinIR (Liang et al., 2021) | 225.09 | 54.07 | 30.56 | 310.32 | 11140.44 |
| Restormer (Zamir et al., 2022) | 139.12 | 74.58 | **30.37** | 244.07 | 2153.51 |
| DLGSANet (Li et al., 2023c) | 142.93 | 50.53 | 30.75 | 224.21 | 7743.70 |
| HAT (Chen et al., 2023) | 266.39 | 54.93 | 164.37 | 485.69 | 1019.07 |
| **ShareFormer (ours)** | **94.45** | **24.57** | 30.61 | **149.63** | **338.79** |

Table 1: Latency and trainability evaluation of lightweight $\times 2$ image SR task on the NVIDIA RTX 3090 GPU. Trainability is assessed using neural tangent kernel condition number (Chen et al., 2021b), where lower values indicate faster convergence. More results with various other devices are given in Appendix A.

To find out the solution to questions (a) and (b), many efficient Transformer-based methods (Zamir et al., 2022; Lu et al., 2022; Yawei Li et al., 2023; Wang et al., 2023) have been proposed. The mainstream approach is to incorporate convolutional operators to augment or substitute certain self-attention modules. These methods leverage the advantage of CNNs in efficiently processing high-frequency information (Li et al., 2023a), introduce appropriate inductive bias (d'Ascoli et al., 2021; Barzilai et al., 2023) for Transformers, and expedite network convergence. However, these methods are not efficient enough while being sufficiently effective. They were insufficient to address the issue (a) stemming from self-attention mechanisms due to their excessive use of matrix multiplication, softmax, and other intricate operators. This assertion was corroborated by us using NVIDIA-DLProf to correlate profile timing data on previous Transformers (Liang et al., 2021; Zamir et al., 2022; Li et al., 2023c; Chen et al., 2023). The results are shown in Tab. 1, which suggests that the key to reducing inference time lies in lowering the computational complexity of self-attention modules in Transformers.

Based on these findings, we share attention maps among neighboring layers to avoid generating attention maps of the network due to their high cost, which is aptly named the **shared portion stripe attention (SPSA)**. Nevertheless, SPSA disrupts the information flow, causing poor optimization of the model. To address this issue while still benefiting from SPSA, which saves almost 33% of the CUDA runtime, we introduce the **residual connection to the "Value"** of self-attention. To validate the necessity and effectiveness of these residual connections, we conducted lesion studies (Veit et al., 2016) to show that SPSA-based Transformers with residuals can be equated to the integration of multiple shallow Transformers with shared attention. This ensemble property introduces local inductive bias (Barzilai et al., 2023) and significantly improves the model's trainability. Furthermore, the **combined shared attention unit (CSAU)**, reduces the computational burden of Feed Forward Network (FFN) by implementing gating units on the shared attention. This unit improves the output properties of SPSA and prevents potential inaccuracies in Shared Attention whilst ensuring trainability. So far, the foundational **share attention block (SAB)** of **ShareFormer** has been constructed using SPSA, complete with residual connections on "value" and CSAU, with the goal of achieving high performance, efficiency, and trainability. To summarize, our contributions are as follows:

- For question(a), we propose SAB to reduce the network latency massively. It achieves up to $7\times$ speedup on high-resolution image denoising compared to previous methods.

- For question(b), we introduce residual connections to the "Value" in SAB, making SAB converge faster. We show that this design offers appropriate local inductive bias into the network without increasing latency.

- We use the SAB to build the ShareFormer, a novel Transformer network. ShareFormer strikes an outstanding balance between latency, trainability, and performance across diverse image restoration tasks.

## 2 RELATED WORK

**Transformers in Image Restoration** Based on Swin Transformer (Liu et al., 2021) and its shifted windows, SwinIR (Liang et al., 2021) was initially proposed to set the new baselines in various image restoration tasks. Subsequently, several Transformer-based methods (Chen et al., 2021a; Wang et al., 2022; Chen et al., 2023; Yawei Li et al., 2023) have been proposed in succession. IPT (Chen et al., 2021a) uses the ImageNet (Russakovsky et al., 2015) to generate a large number of

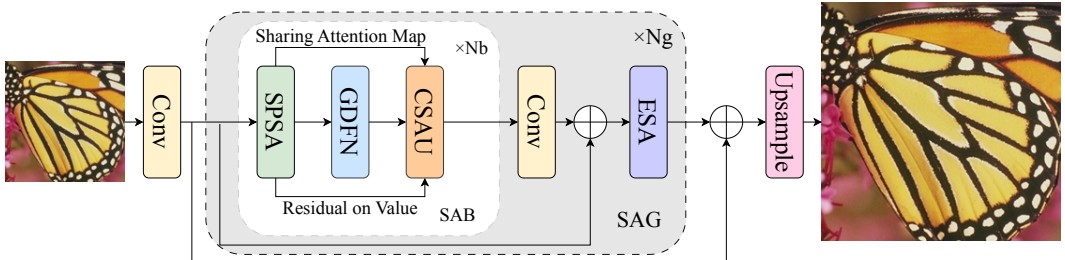

Figure 2: The architecture of ShareFormer for image SR.

corrupted image pairs to pretrain the model for improving the performance. HAT (Chen et al., 2023) found that SwinIR was not able to extract more global information than CNN-based methods such as RCAN (Zhang et al., 2018b), but the mix of channel attention (Vaswani et al., 2017) with overlapping window attention could. Although these methods achieve state-of-the-art (SOTA), they introduce a large amount of computational overhead by computing with heavier attention mechanisms.

**Efficient Attention in Transformer** Due to the high computational costs of the Transformer-based methods, more researchers (Katharopoulos et al., 2020; He et al., 2021; Yang et al., 2021; Hua et al., 2022; Mehta & Rastegari, 2022) are focusing on how to design the architecture with fewer parameters and lower latency for different tasks. Specifically for image restoration tasks, some researchers achieve this through the hybrid networks (Fang et al., 2022; Zhang et al., 2022; Choi et al., 2023; Wang et al., 2023; Li et al., 2023a). They use the addition of convolutional modules to bring a more vigorous representation of high-frequency information and use the self-attention mechanism to ensure that these networks have a high perceptual field. As opposed to this, other methods (Zamir et al., 2022; Zhang et al., 2022; Yawei Li et al., 2023; Li et al., 2023c) decrease the computational resource usage of attention map generation by incorporating information in the global, regional, and local range, instead of just incorporating information globally. They also partition the input features and lessen the attention map's resolution slightly to attain efficient computation during inference.

## 3 METHOD

### 3.1 OVERALL ARCHITECTURE

The overall architecture of the proposed ShareFormer is illustrated in Fig. 2. Here image super-resolution (SR) is presented as a demonstrative task. Given an input image $I \in \mathbb{R}^{H \times W \times 3}$, where $H \times W$ denotes the resolution, ShareFormer firstly applies a convolution layer to obtain shallow feature $F_0 \in \mathbb{R}^{H \times W \times C}$, where $C$ is the embedding dimension of the network. Next, following RCAN (Zhang et al., 2018b), ShareFormer adopts a residual-in-residual structure to construct a deep feature extraction module comprised of $N_g$ share attention groups (SAG). Each SAG consists of $N_b$ share attention blocks (SAB) and a convolution layer. Each SAB, in turn, is composed of SPSA, gated-Dconv feed-forward network (GDFN) (Zamir et al., 2022), CSAU, and enhanced spatial attention (ESA) (Liu et al., 2020b). When the shallow feature $F_0$ is fed into the deep feature extraction module, in each residual group, and within each attention block, SPSA computes the attention map, GDFN performs spatial adjustments, and CASU integrates attention information. Subsequently, a convolutional layer is employed to extract the deep feature $F_1 \in \mathbb{R}^{H \times W \times C}$. After the extraction through the deep feature extraction module, we employ the restoration module to generate the high-quality image $\hat{I}$ from the feature $F_r = F_0 + F_1$. As for other image restoration tasks that do not involve changes in resolution, we build the ShareFormer model following Restormer.

### 3.2 SHARED PORTION STRIPE ATTENTION

In this section, we describe the details of our SPSA and its compatibility with other self-attention methods. Starting from the vanilla attention (VA) and following the above analysis of the efficiency of Transformer-based methods, we will gradually remove the computational overheads in VA to obtain our SPSA module. For the sake of simplicity, let us express the VA as follows:

$$VA(X) = \text{Softmax}(\frac{W_q X (W_k X)^T}{s}) W_v X, \tag{1}$$

where $W_q, W_k, W_v$ are the linear transformation weights for calculating query $(Q)$, key $(K)$, and value $(V)$ from the input $X$, and $s$ is a constant scalar. Note that this equation represents that the number of attention heads is 1, and we also disregard the process to transpose the $Q, K$, and $V$ here.

**Portion Stripe Attention.** Window attention (WA) (Liang et al., 2021) is a particular case of stripe attention(SA) (Shi et al., 2023) due to the fact that the shape of the window could be rectangular rather than square. We assume that the window size is set to $[M_h, M_w]$. In the current case, the computational complexity of SA on a feature $F \in \mathbb{R}^{H \times W \times C}$ can be described as:

$$\Omega(\text{SA}) = 4HWC^2 + 2(M_h M_w)HWC \tag{2}$$

In order to allow the network to minimize computational cost, we divide the features according to the channel dimensions and execute the attention computation in portions. The information flow of the portion stripe attention (PSA) could be formulated as Eq. 4. Supposing the $K$ groups of features $\{F_k \in \mathbb{R}^{H \times W \times C}, k \in K\}$ are equally split, the computational complexity of PSA is:

$$\Omega(\text{PSA}) = 4HWC^2 + \sum_k \frac{2}{K}(M_{hk}M_{wk})HWC \tag{3}$$

**Share Attention.** Based on the preceding analysis, we can deduce that the bottleneck in the computation of $Q, K$, and $V$ by PSA, as well as the generation of attention maps, is predominantly influenced by factors $2C$ and $\sum_k \frac{2}{K}(M_{hk}M_{wk})$, respectively. This arises from the need to maintain a consistent resolution with the input features throughout the computations across multiple attention blocks. Hence, a promising strategy for reducing computational complexity naturally emerges, which is to reduce redundant calculations along the information flow between adjacent blocks by sharing $Q, K$, and $V$ pairs. This results in the information flow to be transformed into the Eq. 5.

$$
\begin{aligned}
Q_{l-1}, K_{l-1} &= W_q(X_{l-1}), W_k(X_{l-1}); \\
Attn_{l-1} &= Softmax((Q_{l-1}K_{l-1}^T)/s); \\
V_{l-1} &= W_v(X_{l-1}); \\
X_l &= W_{l-1}(Attn_{l-1}V_{l-1}) + X_{l-1}; \\
X_l &= FFN(X_l); \\
Q_l, K_l, V_l &= W_q(X_l), W_k(X_l), W_v(X_l); \\
Attn_l &= Softmax((Q_l K_l^T)/s); \\
X_{l+1} &= FFN(W_l(Attn_l V_l) + X_l);
\end{aligned}
\tag{4}
$$

$$
\begin{aligned}
Q_{l-1}, K_{l-1} &= W_q(X_{l-1}), W_k(X_{l-1}); \\
Attn_{l-1} &= Softmax((Q_{l-1}K_{l-1}^T)/s); \\
V_{l-1} &= W_v(X_{l-1}); \\
X_l &= W_{l-1}(Attn_{l-1}V_{l-1}) + X_{l-1}; \\
X_l &= FFN(X_l); \\
\mathbf{G_l}, V_l &= \mathbf{W_g X_l}, W_v(X_l) + V_{l-1}; \\
Attn_l &= Attn_{l-1}; \\
X_{l+1} &= W_l(\mathbf{G_l} \odot (\mathbf{Attn_l V_l})) + X_l;
\end{aligned}
\tag{5}
$$

where $Attn_l$ denotes the attention map of the block $l$, $\odot$ denotes element-wise multiplication. Shared attention calculation is highlighted in red, V through the residual connection is highlighted in blue, and the gate unit is **bolded**.

The primary motivation for SPSA is to decrease the coefficient of the first term of Eq. 3 by reducing the number of convolutions. Additionally, by multiplexing the attention maps, we aim to decrease the coefficient of the second term of Eq. 3, ultimately resulting in a significant acceleration of the attention mechanism. At this point, the computational complexity of SPSA could be decreased to Eq. 6. In comparison to Eq. 3, the computational overhead extruded by Eq. 6 is in fact redundant calculations, as supported by the corresponding empirical evidence provided in Appendix D.

$$
\begin{aligned}
\Omega(SPSA) &= 0.5(\overbrace{4HWC^2 + \sum_k \frac{2}{K}(M_{hk}M_{wk})HWC}^{\text{computational complexity of block } l-1} + \overbrace{2HWC^2 + \sum_k \frac{1}{K}(M_{hk}M_{wk})HWC}^{\text{computational complexity of block } l}) \\
&= 3HWC^2 + \sum_k \frac{3}{2K}(M_{hk}M_{wk})HWC
\end{aligned}
\tag{6}
$$

**Residual Connections on Value.** While utilizing the shared attention offers a substantial speed increase, it hinders the regular flow of information throughout the network. In order to offset this

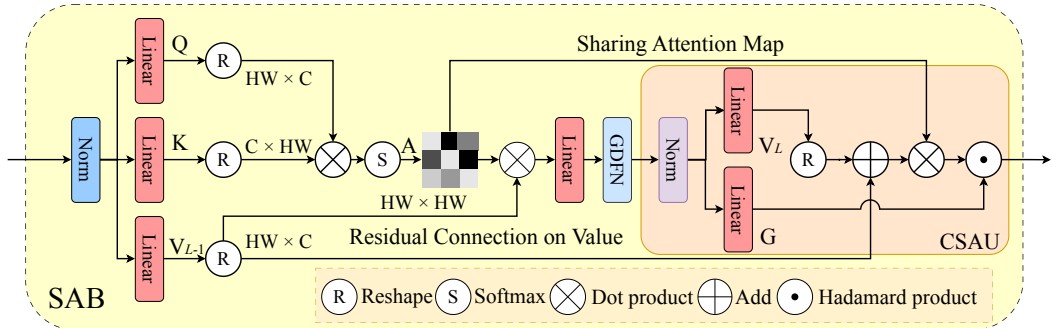

Figure 3: Schematic of the structure of the SAB.

deficiency, we establish residual connections on $V$ between the neighboring attention blocks. So far, the final architecture of SPSA can be observed in Fig. 3. We shall dedicate Sec. 4 to examining how this simple operation can introduce inductive bias to the network, along with an explanation of the underlying mechanisms.

### 3.3 COMBINE SPSA WITH GATED UNIT

Unfortunately, the outputs of the shared attention module may contain noise, while gating is an efficient strategy for regulating information. So in order to better control the output distribution of the SPSA, we use the GDFN (Zamir et al., 2022) as a basic module of our SAB. To further reduce the computational complexity, we propose the CSAU to combine the SPSA and GDFN into a cohesive layer defined by the Eq. 5. Furthermore, incorporating gate units and shared attention also alleviates the significant computational burden associated with directly implementing FFN.

## 4 TRAINABILITY

In this section, we substantiate the improvement in model trainability resulting from the incorporation of residual connections on the "Value" of SPSA through a series of lesion studies. These studies effectively align our model with an ensemble of multiple shallow Transformers with shared attention modules. As a consequence, the model gains a denser receptive field which leads to some locality bias that enhances the overall trainability.

### 4.1 ENSEMBLES OF SHARED ATTENTIONS

**Lesion Study.** We conducted experiments based on a 72-layer ShareFormer trained on $4\times$ SR task. Following Veit et al. (2016), we delete or reorder residual modules in ShareFormer during test. Fig. 4 and Fig. 5 show that MAE loss increases smoothly when randomly deleting or ordering several modules in ShareFormer. These results closely align with the expected outcomes of ensemble.

It should be noted that shared attention separates information flow from residual branches, resulting in the model's information pathway consisting of only one path. Hence, Transformers with shared attention, but lacking residual values, cannot be regarded as the ensembles of shallow networks.

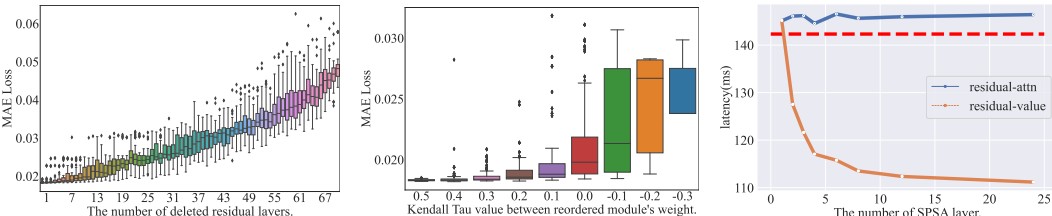

Figure 4: MAE loss increases smoothly when randomly deleting modules.

Figure 5: MAE loss increases smoothly when randomly reordering modules.

Figure 6: Residual connections on the attention maps do not result in acceleration.

**Residual Connections on Values vs. on Attention Maps.** Since adding residual connections to attention maps does not reduce the number of softmax operations or convolutions in the model, placing

residuals on the attention maps will not speed up the network, particularly when the share number increases, see Fig. 6. Furthermore, this will cause the attention maps to become homogeneous.

## 4.2 EFFECTIVE RECEPTIVE FIELD AND LOCALITY BIAS

The preceding lesion study has unveiled that our ShareFormer fundamentally equates to an ensemble of multiple shallow Transformers with shared attention. In essence, the efficient receptive field (ERF) of the ShareFormer is equivalent to the weighted integration of ERFs of the multiple shallow Transformers. As elucidated by Barzilai et al. (2023), the ensemble behavior of the network results in its ERF carrying more weight towards the central region of the receptive field. This observation implies an amplified locality bias of the network, which has been proven to be capable of facilitating the training (d'Ascoli et al., 2021).

We visualize the ERF of two ShareFormer models in Fig. 7: the left one lacks residual connections on "Value", while the right one includes them. The visualization indicates that the network incorporating residual connections on "Value" exhibits a more pronounced concentration of weight at the central area of the ERF. This observation suggests that the inclusion of residual connections on "Value" of ShareFormer introduces a locality bias, consequently enhancing the network's trainability. Fig. 7 also includes PSNR and neural tangent kernel condition numbers (Chen et al., 2021b), demonstrating improved both trainability and performance with the introduction of residual "Value," effectively resolving the original trade-off.

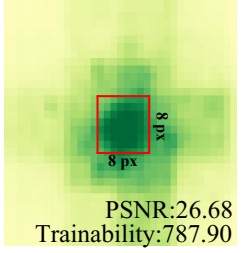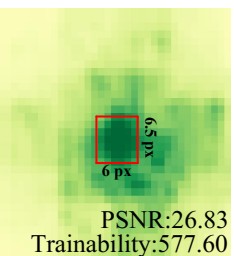

PSNR:26.68
Trainability:787.90

PSNR:26.83
Trainability:577.60

Figure 7: Residual connections on "Value" brings more concentrated ERF. Trainability is assessed by the neural tangent kernel condition number (Chen et al., 2021b), with smaller values indicating higher trainability. (Left / Right: without / with residual "Value")

## 5 EXPERIMENT RESULTS

We evaluate the ShareFormer on three tasks: **image SR**, **image denoising**, and **JPEG compression artifact reduction (CAR)**. To avoid problems such as gradient vanishing, gradient explosion, and overfitting, we utilize Smooth $L_1$ loss (Huber, 1992) for image SR and Charbonnier loss (Charbonnier et al., 1994) for image denoising and JPEG CAR. The training datasets and protocols, implementation details, and additional results are presented in Appendix B-C. The latency reported is averaged over 1000 repetitions with an NVIDIA RTX 3090 GPU on $1280 \times 720$ resolution. "N/A" indicates that the corresponding models are too heavy for the NVIDIA RTX 3090 GPU. When self-ensemble strategy (Lim et al., 2017) is used in testing, we mark the model with a symbol "+".

### 5.1 RESULTS ON IMAGE SR

#### 5.1.1 CLASSICAL IMAGE SR

**Quantitative comparison** between ShareFormer and several SOTA methods including RCAN (Zhang et al., 2018b), DRLN (Anwar & Barnes, 2020), HAN (Niu et al., 2020), NLSA (Mei et al., 2021), SwinIR (Liang et al., 2021), ELAN (Zhang et al., 2022), EDT (Li et al., 2023b), and ART (Zhang et al., 2023), is reported in Table 2. As we can see, ShareFormer achieves the best performance on almost all five benchmark datasets for all scale factors. In particular, ShareFormer archives a PSNR score of 34.19dB on the Urban100 dataset, surpassing SwinIR by 0.38dB, while maintaining only 62% parameters and delivering over 2 times faster inference speed. When compared to CNN-based methods, while it is hard to achieve a shorter inference time, ShareFormer exhibits superior performance due to capturing long-range feature dependencies through SPSA.

**Qualitative comparison.** We present challenging examples for qualitative comparison ($\times 4$) in Fig. 8. It is evident that ShareFormer prevents the output image from being too sharp by reducing the number of self-attention maps produced. This results in a more appealing visual effect.

#### 5.1.2 LIGHTWEIGHT IMAGE SR

**Quantitative comparison** between ShareFormer and several SOTA methods including IMDN (Hui et al., 2019), RFDN (Liu et al., 2020a), SwinIR (Liang et al., 2021), ELAN (Zhang et al., 2022), and DLGSA (Li et al., 2023c), is reported in Table 3. It can be seen that ShareFormer-L performs significantly better on high-resolution datasets such as Urban100 and Manga109. Interestingly, we

| Method | Scale | Params (M) | FLOPs (T) | Latency (ms) | Set5 PSNR / SSIM | Set14 PSNR / SSIM | BSD100 PSNR / SSIM | Urban100 PSNR / SSIM | Manga109 PSNR / SSIM |
|---|---|---|---|---|---|---|---|---|---|
| RCAN (Zhang et al., 2018b) | | 15.44 | 3.77 | 585.03 | 38.27 / 0.9614 | 34.12 / 0.9216 | 32.41 / 0.9027 | 33.34 / 0.9384 | 39.44 / 0.9786 |
| DRLN (Anwar & Barnes, 2020) | | 15.44 | 8.39 | 731.98 | 38.27 / 0.9616 | 34.28 / 0.9231 | 32.44 / 0.9028 | 33.37 / 0.9390 | 39.58 / 0.9792 |
| HAN (Niu et al., 2020) | | 63.61 | 15.52 | 1441.23 | 38.27 / 0.9614 | 34.16 / 0.9217 | 32.41 / 0.9027 | 33.35 / 0.9385 | 39.46 / 0.9787 |
| NLSA (Mei et al., 2021) | | 41.80 | 10.27 | 1345.97 | 38.34 / 0.9618 | 34.08 / 0.9231 | 32.43 / 0.9027 | 33.42 / 0.9394 | 39.59 / 0.9789 |
| SwinIR (Liang et al., 2021) | ×2 | 11.75 | 2.99 | 1340.49 | 38.42 / 0.9623 | 34.46 / 0.9250 | 32.53 / 0.9041 | 33.81 / 0.9427 | 39.92 / 0.9797 |
| ELAN* (Zhang et al., 2022) | | 8.25 | 2.02 | 1039.02 | 38.36 / 0.9620 | 34.20 / 0.9228 | 32.45 / 0.9030 | 33.44 / 0.9391 | 39.62 / 0.9793 |
| EDT (Li et al., 2023b) | | 11.48 | 2.84 | 2439.41 | 38.45 / 0.9624 | 34.57 / 0.9258 | 32.52 / 0.9041 | 33.80 / 0.9425 | 39.93 / 0.9800 |
| ART-S (Zhang et al., 2023) | | 11.72 | 3.49 | N/A | 38.48 / 0.9625 | 34.50 / 0.9258 | 32.53 / 0.9043 | 34.02 / 0.9437 | 40.11 / 0.9804 |
| **ShareFormer (Ours)** | | **7.52** | **1.76** | **939.22** | **38.53 / 0.9626** | **34.62 / 0.9260** | **32.59 / 0.9049** | **34.22 / 0.9451** | **40.10 / 0.9801** |
| **ShareFormer+ (Ours)** | | **7.52** | **1.76** | **939.22** | **38.57 / 0.9627** | **34.69 / 0.9264** | **32.62 / 0.9052** | **34.40 / 0.9462** | **40.20 / 0.9803** |
| RCAN (Zhang et al., 2018b) | | 15.63 | 1.78 | 283.36 | 34.74 / 0.9299 | 30.65 / 0.8482 | 29.32 / 0.8111 | 29.09 / 0.8702 | 34.44 / 0.9499 |
| DRLN (Anwar & Barnes, 2020) | | 34.61 | 3.94 | 353.05 | 34.78 / 0.9303 | 30.73 / 0.8488 | 29.36 / 0.8117 | 29.21 / 0.8722 | 34.71 / 0.9509 |
| HAN (Niu et al., 2020) | | 64.35 | 7.33 | 686.49 | 34.75 / 0.9299 | 30.67 / 0.8483 | 29.32 / 0.8110 | 29.10 / 0.8705 | 34.48 / 0.9500 |
| NLSA (Mei et al., 2021) | | 44.75 | 5.14 | 657.33 | 34.85 / 0.9306 | 30.70 / 0.8485 | 29.34 / 0.8117 | 29.25 / 0.8726 | 34.57 / 0.9508 |
| SwinIR (Liang et al., 2021) | ×3 | 11.94 | 1.42 | 605.45 | 34.97 / 0.9318 | 30.93 / 0.8534 | 29.46 / 0.8145 | 29.75 / 0.8826 | 35.12 / 0.9537 |
| ELAN* (Zhang et al., 2022) | | 8.28 | 0.94 | 456.57 | 34.90 / 0.9313 | 30.80 / 0.8504 | 29.38 / 0.8124 | 29.32 / 0.8745 | 34.73 / 0.9517 |
| EDT (Li et al., 2023b) | | 11.66 | 1.19 | 993.25 | 34.97 / 0.9316 | 30.89 / 0.8527 | 29.44 / 0.8142 | 29.72 / 0.8814 | 35.13 / 0.9534 |
| ART-S (Zhang et al., 2023) | | 11.90 | 1.66 | 1682.21 | 34.98 / 0.9318 | 30.94 / 0.8530 | 29.45 / 0.8146 | 29.86 / 0.8830 | 35.22 / 0.9539 |
| **ShareFormer (Ours)** | | **7.71** | **0.85** | **422.62** | **34.99 / 0.9323** | **30.96 / 0.8537** | **29.50 / 0.8158** | **29.95 / 0.8851** | **35.32 / 0.9543** |
| **ShareFormer+ (Ours)** | | **7.71** | **0.85** | **422.62** | **35.08 / 0.9326** | **31.07 / 0.8548** | **29.53 / 0.8164** | **30.14 / 0.8875** | **35.49 / 0.9549** |
| RCAN (Zhang et al., 2018b) | | 15.59 | 0.10 | 178.01 | 32.63 / 0.9002 | 28.87 / 0.7889 | 27.77 / 0.7436 | 26.82 / 0.8087 | 31.22 / 0.9173 |
| DRLN (Anwar & Barnes, 2020) | | 34.58 | 0.21 | 202.62 | 32.63 / 0.9002 | 28.94 / 0.7900 | 27.83 / 0.7444 | 26.98 / 0.8119 | 31.54 / 0.9196 |
| HAN (Niu et al., 2020) | | 64.20 | 0.40 | 388.71 | 32.64 / 0.9002 | 28.90 / 0.7890 | 27.80 / 0.7442 | 26.85 / 0.8094 | 31.42 / 0.9177 |
| NLSA (Mei et al., 2021) | | 44.16 | 0.32 | 377.51 | 32.59 / 0.9000 | 28.87 / 0.7891 | 27.78 / 0.7444 | 26.96 / 0.8109 | 31.27 / 0.9184 |
| SwinIR (Liang et al., 2021) | ×4 | 11.90 | 0.08 | 293.25 | 32.92 / 0.9044 | 29.09 / 0.7950 | 27.92 / 0.7489 | 27.45 / 0.8254 | 32.03 / 0.9260 |
| ELAN* (Zhang et al., 2022) | | 8.31 | 0.05 | 236.56 | 32.75 / 0.9022 | 28.96 / 0.7914 | 27.83 / 0.7459 | 27.13 / 0.8167 | 31.68 / 0.9226 |
| EDT (Li et al., 2023b) | | 11.63 | 0.07 | 590.26 | 32.82 / 0.9031 | 29.09 / 0.7939 | 27.91 / 0.7483 | 27.46 / 0.8246 | 32.03 / 0.9254 |
| ART-S (Zhang et al., 2023) | | 11.87 | 0.09 | 547.52 | 32.86 / 0.9029 | 29.09 / 0.7942 | 27.91 / 0.7489 | 27.54 / 0.8261 | 32.13 / 0.9263 |
| **ShareFormer (Ours)** | | **7.67** | **0.05** | **211.10** | **32.83 / 0.9033** | **29.11 / 0.7950** | **27.94 / 0.7498** | **27.64 / 0.8294** | **32.18 / 0.9262** |
| **ShareFormer+ (Ours)** | | **7.67** | **0.05** | **211.10** | **32.97 / 0.9038** | **29.20 / 0.7960** | **27.98 / 0.7506** | **27.82 / 0.8324** | **32.41 / 0.9278** |

Table 2: Quantitative comparison (PSNR/SSIM) for ***classical image SR*** with the SOTA methods on benchmark datasets. The best and second-best results among Transformer-based methods are marked in red and blue, respectively. The CNN-based methods and Transformer-based methods are separated via a dashed line for each scaling factor. "*" means the model was trained with the DIV2K dataset.

| Method | Scale | Params (K) | FLOPs (G) | Latency (ms) | Set5 PSNR / SSIM | Set14 PSNR / SSIM | BSD100 PSNR / SSIM | Urban100 PSNR / SSIM | Manga109 PSNR / SSIM |
|---|---|---|---|---|---|---|---|---|---|
| IMDN (Hui et al., 2019) | | 694 | 162.58 | 39.71 | 38.00 / 0.9605 | 33.63 / 0.9177 | 32.19 / 0.8996 | 32.17 / 0.9283 | 38.88 / 0.9774 |
| RFDN-L (Liu et al., 2020a) | | 626 | 145.32 | 45.24 | 38.08 / 0.9606 | 33.67 / 0.9190 | 32.18 / 0.8996 | 32.24 / 0.9290 | 38.95 / 0.9773 |
| SwinIR-L (Liang et al., 2021) | ×2 | 910 | 240.72 | 310.32 | 38.14 / 0.9611 | 33.86 / 0.9206 | 32.31 / 0.9012 | 32.76 / 0.9340 | 39.12 / 0.9783 |
| ELAN (Zhang et al., 2022) | | 582 | 135.88 | 192.44 | 38.17 / 0.9611 | 33.94 / 0.9207 | 32.30 / 0.9012 | 32.76 / 0.9340 | 39.11 / 0.9782 |
| DLGSA-L (Li et al., 2023c) | | 745 | 173.96 | 224.21 | 38.20 / 0.9612 | 33.89 / 0.9203 | 32.30 / 0.9012 | 32.94 / 0.9355 | 39.29 / 0.9780 |
| **ShareFormer-L (Ours)** | | **535** | **117.50** | **149.63** | **38.27 / 0.9615** | **34.13 / 0.9232** | **32.38 / 0.9023** | **33.13 / 0.9373** | **39.46 / 0.9787** |
| IMDN (Hui et al., 2019) | | 703 | 72.47 | 15.34 | 34.36 / 0.9270 | 30.32 / 0.8417 | 29.09 / 0.8046 | 28.17 / 0.8519 | 33.61 / 0.9445 |
| RFDN-L (Liu et al., 2020a) | | 633 | 64.70 | 17.17 | 34.47 / 0.9280 | 30.35 / 0.8421 | 29.11 / 0.8053 | 28.32 / 0.8547 | 33.78 / 0.9458 |
| SwinIR-L (Liang et al., 2021) | ×3 | 918 | 105.97 | 134.52 | 34.62 / 0.9289 | 30.54 / 0.8463 | 29.20 / 0.8082 | 28.66 / 0.8624 | 33.98 / 0.9478 |
| ELAN (Zhang et al., 2022) | | 590 | 60.66 | 84.94 | 34.61 / 0.9288 | 30.55 / 0.8463 | 29.21 / 0.8081 | 28.69 / 0.8624 | 34.00 / 0.9478 |
| DLGSA-L (Li et al., 2023c) | | 752 | 76.52 | 102.31 | 34.70 / 0.9295 | 30.58 / 0.8465 | 29.24 / 0.8089 | 28.83 / 0.8653 | 34.16 / 0.9483 |
| **ShareFormer-L (Ours)** | | **543** | **52.57** | **67.51** | **34.73 / 0.9299** | **30.66 / 0.8485** | **29.30 / 0.8106** | **29.01 / 0.8692** | **34.54 / 0.9501** |
| IMDN (Hui et al., 2019) | | 715 | 43.69 | 11.21 | 32.21 / 0.8948 | 28.58 / 0.7811 | 27.56 / 0.7353 | 26.04 / 0.7838 | 30.45 / 0.9075 |
| RFDN-L (Liu et al., 2020a) | | 643 | 38.95 | 12.60 | 32.28 / 0.8957 | 28.61 / 0.7818 | 27.58 / 0.7363 | 26.20 / 0.7883 | 30.61 / 0.9096 |
| SwinIR-L (Liang et al., 2021) | ×4 | 930 | 62.80 | 78.20 | 32.44 / 0.8976 | 28.77 / 0.7858 | 27.69 / 0.7406 | 26.47 / 0.7980 | 30.92 / 0.9151 |
| ELAN (Zhang et al., 2022) | | 601 | 36.65 | 50.69 | 32.43 / 0.8975 | 28.78 / 0.7858 | 27.69 / 0.7406 | 26.54 / 0.7982 | 30.92 / 0.9150 |
| DLGSA-L (Li et al., 2023c) | | 761 | 45.51 | 86.33 | 32.54 / 0.8993 | 28.84 / 0.7871 | 27.73 / 0.7415 | 26.66 / 0.8033 | 31.13 / 0.9161 |
| **ShareFormer-L (Ours)** | | **555** | **31.85** | **41.47** | **32.54 / 0.8993** | **28.94 / 0.7893** | **27.77 / 0.7434** | **26.83 / 0.8080** | **31.40 / 0.9189** |

Table 3: Quantitative comparison for ***lightweight image SR*** with SOTA methods.

observe that ShareFormer exhibits a relatively larger advantage compared to other methods under a reduced parameter regime, suggesting that SPSA may be particularly well-suited for lightweight neural networks. For lightweight ×2 image SR, ShareFormer-L has increased 0.34dB than SwinIR on the Manga109 dataset, but there is only 0.21dB increase for classical ×2 image SR. This further leads to the conjecture that lightweight Transformers may have more homogeneous attention maps, so cross-layer sharing of attention maps could help the model perform better. Meanwhile, the homogeneity of attention maps may be the key that limits the ability of lightweight Transformers.

**Qualitative comparison.** As shown in Fig. 8, ShareFormer-L is the only method that precisely handles the densely packed small structures in the image, such as the windows of the building.

### 5.2 RESULT ON IMAGE DENOISING

Table 4 and Table 5 show the quantitative comparisons between ShareFormer and representative methods including RNAN (Zhang et al., 2019), BRDNet (Tian et al., 2020), IPT (Chen et al., 2021a),

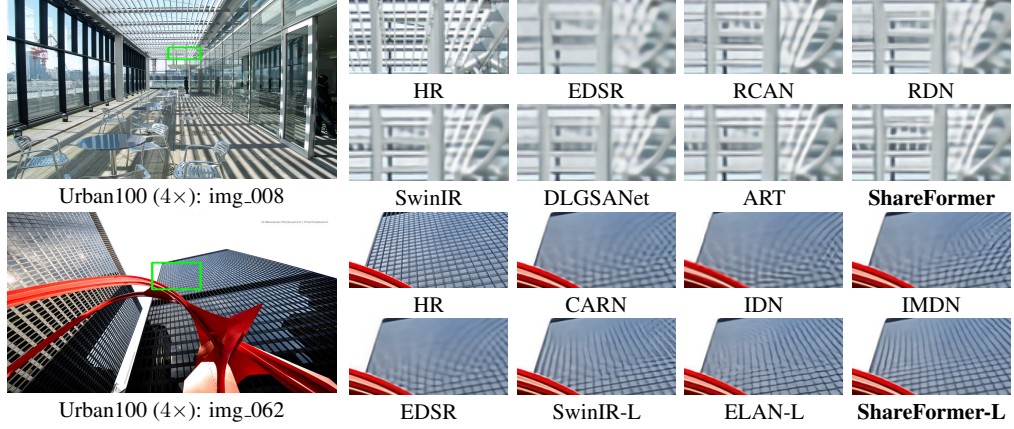

Figure 8: Qualitative comparison with recent SOTA methods on the 4× *image SR* task. The top row is **classical image SR** results, and the bottom row is **lightweight image SR** results.

| Method | Latency | Set12 | | | BSD68 | | | Urban100 | | |
|---|---|---|---|---|---|---|---|---|---|---|
| | (ms) | $\sigma$=15 | $\sigma$=25 | $\sigma$=50 | $\sigma$=15 | $\sigma$=25 | $\sigma$=50 | $\sigma$=15 | $\sigma$=25 | $\sigma$=50 |
| DnCNN (Zhang et al., 2017) | 83.18 | 32.67 | 30.35 | 27.18 | 31.62 | 29.16 | 26.23 | 32.28 | 29.80 | 26.35 |
| DRUNet (Zhang et al., 2021) | 172.39 | 33.25 | 30.94 | 27.90 | 31.91 | 29.48 | 26.59 | 33.44 | 31.11 | 27.96 |
| Restormer (Zamir et al., 2022) | 1201.53 | 33.35 | 31.04 | 28.01 | 31.95 | 29.51 | 26.62 | 33.67 | 31.39 | 28.33 |
| **ShareFormer (Ours)** | 806.97 | 33.34 | 31.02 | 27.97 | 31.96 | 29.52 | 26.62 | 33.71 | 31.44 | 28.36 |
| RNAN (Zhang et al., 2019) | N/A | - | - | 27.70 | - | - | 26.48 | - | - | 27.65 |
| DeamNet (Ren et al., 2021) | N/A | 33.19 | 30.81 | 27.74 | 31.91 | 29.44 | 26.54 | 33.37 | 30.85 | 27.53 |
| SwinIR (Liang et al., 2021) | 5689.86 | 33.36 | 31.01 | 27.91 | 31.97 | 29.50 | 26.58 | 33.70 | 31.30 | 27.98 |
| Restormer (Zamir et al., 2022) | 1201.53 | 33.42 | 31.08 | 28.00 | 31.96 | 29.52 | 26.62 | 33.79 | 31.46 | 28.29 |
| **ShareFormer (Ours)** | 806.97 | 33.42 | 31.06 | 27.98 | 31.99 | 29.53 | 26.62 | 33.86 | 31.55 | 28.37 |
| **ShareFormer+ (Ours)** | 806.97 | 33.44 | 31.09 | 28.02 | 32.00 | 29.55 | 26.64 | 33.92 | 31.63 | 28.46 |

Table 4: Quantitative comparison for *grayscale image denoising*. Top row: $\sigma$-mixed training; Bottom row: $\sigma$-specific training.

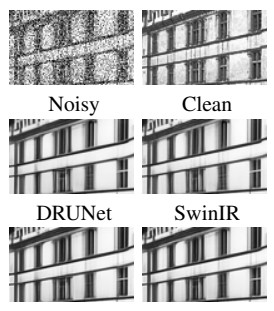

Figure 9: *Grayscale image denoising* results.

| Method | CBSD68 | | | Kodak24 | | | McMaster | | | Urban100 | | |
|---|---|---|---|---|---|---|---|---|---|---|---|---|
| | $\sigma$=15 | $\sigma$=25 | $\sigma$=50 | $\sigma$=15 | $\sigma$=25 | $\sigma$=50 | $\sigma$=15 | $\sigma$=25 | $\sigma$=50 | $\sigma$=15 | $\sigma$=25 | $\sigma$=50 |
| DnCNN (Zhang et al., 2017) | 33.90 | 31.24 | 27.95 | 34.60 | 32.14 | 28.95 | 33.45 | 31.52 | 28.62 | 32.98 | 30.81 | 27.59 |
| DRUNet (Zhang et al., 2021) | 34.30 | 31.69 | 28.51 | 35.31 | 32.89 | 29.86 | 35.40 | 33.14 | 30.08 | 34.81 | 32.60 | 29.61 |
| Restormer (Zamir et al., 2022) | 34.39 | 31.78 | 28.59 | 35.44 | 33.02 | 30.00 | 35.55 | 33.31 | 30.29 | 35.06 | 32.91 | 30.02 |
| **ShareFormer (Ours)** | 34.41 | 31.79 | 28.59 | 35.48 | 33.04 | 30.00 | 35.57 | 33.31 | 30.26 | 35.13 | 32.98 | 30.07 |
| RNAN (Zhang et al., 2019) | - | - | 28.27 | - | - | 29.58 | - | - | 29.72 | - | - | 29.08 |
| BRDNet (Tian et al., 2020) | 34.10 | 31.43 | 28.16 | 34.88 | 32.41 | 29.22 | 35.08 | 32.75 | 29.52 | 34.42 | 31.99 | 28.56 |
| IPT (Chen et al., 2021a) | - | - | 28.39 | - | - | 29.64 | - | - | 29.98 | - | - | 29.71 |
| SwinIR (Liang et al., 2021) | 34.42 | 31.78 | 28.56 | 35.34 | 32.89 | 29.79 | 35.61 | 33.20 | 30.22 | 35.13 | 32.90 | 29.82 |
| Restormer (Zamir et al., 2022) | 34.40 | 31.79 | 28.60 | 35.47 | 33.04 | 30.01 | 35.61 | 33.34 | 30.30 | 35.13 | 32.96 | 30.02 |
| EDT-B (Li et al., 2023b) | 34.39 | 31.76 | 28.56 | 35.37 | 32.94 | 29.87 | 35.61 | 33.34 | 30.25 | 35.22 | 33.07 | 30.16 |
| **ShareFormer (Ours)** | 34.43 | 31.80 | 28.60 | 35.50 | 33.07 | 30.01 | 35.62 | 33.35 | 30.27 | 35.20 | 33.05 | 30.10 |
| **ShareFormer+ (Ours)** | 34.45 | 31.82 | 28.62 | 35.53 | 33.09 | 30.04 | 35.66 | 33.39 | 30.31 | 35.26 | 33.12 | 30.20 |

Table 5: Quantitative comparison for *color image denoising*. Tab. 4 has presented a comparison of the latency.

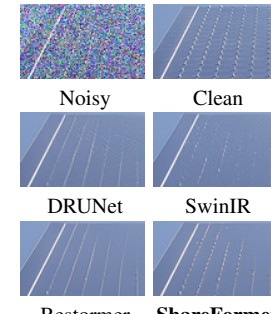

Figure 10: *Color image denoising* results.

DRUNet (Zhang et al., 2021), SwinIR (Liang et al., 2021), Restormer (Zamir et al., 2022), and EDT (Li et al., 2023a) on image denoising. The compared noise levels were set to 15, 25, and 50.

It is evident that our ShareFormer exhibits superior performance in comparison to other methods employed. Furthermore, in comparison to other Transformer-based methods, ShareFormer delivers competitive performance while maintaining the fastest inference speed, courtesy of SPSA. Notably, previous methods that compute attention maps from the spatial perspective, such as SwinIR (Liang et al., 2021), face challenges when applied to high-resolution images due to their computationally intensive nature. Conversely, methods that calculate attention maps from the channel perspective, like Restormer (Zamir et al., 2022), exhibit faster speed but grapple with limited receptive fields. Our ShareFormer, in contrast, boasts a significantly broader receptive field and demonstrates considerably reduced inference latency, effectively mitigating the constraints associated with existing network architectures for high-resolution image restoration.

| Dataset | q | RDN | | DRUNet | | SwinIR | | ART | | ShareFormer | | ShareFormer+ | |
|---|---|---|---|---|---|---|---|---|---|---|---|---|---|
| | | PSNR | SSIM | PSNR | SSIM | PSNR | SSIM | PSNR | SSIM | PSNR | SSIM | PSNR | SSIM |
| **Classic5** | 10 | 30.00 | 0.8188 | 30.16 | 0.8234 | 30.27 | 0.8249 | 30.27 | 0.8258 | 30.28 | 0.8260 | 30.31 | 0.8263 |
| | 30 | 33.43 | 0.8930 | 33.59 | 0.8949 | 33.73 | 0.8961 | 33.74 | 0.8964 | 33.73 | 0.8967 | 33.77 | 0.8969 |
| | 40 | 34.27 | 0.9061 | 34.41 | 0.9075 | 34.52 | 0.9082 | 34.55 | 0.9086 | 34.54 | 0.9089 | 34.57 | 0.9092 |
| **LIVE1** | 10 | 29.67 | 0.8247 | 29.79 | 0.8278 | 29.86 | 0.8287 | 29.89 | 0.8300 | 29.86 | 0.8315 | 29.89 | 0.8321 |
| | 30 | 33.51 | 0.9153 | 33.59 | 0.9166 | 33.69 | 0.9174 | 33.71 | 0.9178 | 33.68 | 0.9190 | 33.71 | 0.9192 |
| | 40 | 34.51 | 0.9302 | 34.58 | 0.9312 | 34.67 | 0.9317 | 34.70 | 0.9322 | 34.66 | 0.9331 | 34.69 | 0.9334 |

Table 6: Quantitative comparison for *JPEG CAR* with SOTA methods on benchmark datasets.

| | Params | Latency | Performance |
|---|---|---|---|
| SPSA+MLP | 9.43M | 238.47 | 27.59 / 0.8275 |
| SPSA+GDFN | 8.91M | 249.32 | 27.61 / 0.8280 |
| CSAU (ours) | 7.67M | 211.10 | 27.64 / 0.8294 |

| Layers | PSNR | SSIM | Latency |
|---|---|---|---|
| 2 | 27.64 | 0.8294 | 211.10 |
| 4 | 27.48 | 0.8241 | 198.44 |
| 6 | 27.45 | 0.8234 | 183.94 |
| RCAN | 26.82 | 0.8087 | 178.01 |

| | MLP | ShiftMLP | GateMLP |
|---|---|---|---|
| WA | 27.59 | 27.59 | 27.62 |
| MHDA | 27.45 | 27.47 | 27.48 |
| SparseGSA | 27.46 | 27.47 | 27.48 |
| PSA (ours) | 27.60 | 27.63 | 27.64 |

Table 7: Ablation experiments for the **CSAU**. It works better and faster with fewer parameters.

Table 8: The impact of **different numbers of shared attention layers.**

Table 9: Performance of Share-Formers with **different attention mechanisms and FFNs.**

Fig. 9 and Fig. 10 present denoised results by different methods for grayscale denoising and color image denoising, respectively. Our ShareFormer restores the cleanest and most faithful image.

## 5.3 RESULT ON JPEG CAR

Tab. 6 shows our ShareFormer obtains satisfactory performance for JPEG CAR on grayscale images. It shows that the ShareFormer effectively restores the texture of images without JPEG compression.

## 5.4 ABLATION STUDY AND DISCUSSION

Here the models are trained for ($\times 4$) image SR task. The results are evaluated on Urban100 dataset. **Impact of combined shared attention unit.** As shown in Tab. 7, when compared with using only SPSA and FFN, CSAU shows a slight improvement in performance and more significant advantages in terms of inference speed and parameters.

**Different Attentions and FFNs.** Eq. 5 shows that our method is compatible with any attention mechanisms, including WA (Liang et al., 2021), MHDA (Zamir et al., 2022) and SparseGSA (Li et al., 2023c). We evaluated the ShareFormer-L using various attentions and FFNs, as presented in Tab. 9. These findings indicate that our shared attention mechanism can be extended to various window attentions and their upgrades, not solely PSA.

**Can Transformer be faster than CNN?** As shown in Tab. 3, SPSA exhibits speedup ratios exceeding $2\times$ when compared to other lightweight Transformers. Tab. 8 provides an overview of ShareFormer's performance and latency across different numbers of shared layers. Notably, Share-Former with 6 shared layers achieves higher accuracy than CNNs like RCAN (Zhang et al., 2018b), while maintaining comparable latency. This finding underscores the potential of Transformer-based methods to offer both superior performance and faster inference times than CNN-based methods.

## 6 CONCLUSION

In this work, we propose ShareFormer for efficient image restoration, which simultaneously achieves the low latency and high trainability of transformers while maintaining peak performance. Most previous methods for image restoration introduce an inductive bias into the network through the incorporation of a convolutional module. However, this approach often led to a notable reduction in inference speed. On the other hand, sparse attention mechanisms were explored to improve speed, but they also introduced challenges related to model training. To address these problems, we devised a strategy named SPSA that involves the sharing of attention maps across adjacent layers, resulting in a substantial reduction in model latency. Additionally, we introduced residual connections to the "Value", resulting in a locality bias and enhancing the model's trainability. Experiments on image SR, denoising, and JPEG CAR validate that our ShareFormer achieves SOTA on various benchmark datasets. In future works, we will apply our ShareFormer to more image restoration tasks, such as image dehazing, deraining, and deblurring. We will also explore the potential of shared attention in solving high-level vision problems, including image classification and segmentation.

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

# APPENDICES

## A LATENCY MEASUREMENT

In the main paper, we report the results of latency measurement of many well-known network structures on NVIDIA RTX 3090 GPU (with constant 100% GPU utilization), due to its wide enough usage. At the same time, we report the latency report of RTX2080TI and RTX4090 in the appendix to prove that our method is valid on many hardware platforms. Since the latency will be affected by the AI system and hardware, to be fair, we unify the normalization and activation layer of the network as LayerNorm and GELU. We do not use CuDNN for acceleration when computing latency.

| Method | RTX3090 (ms) | RTX4090 (ms) | Denoising PSNR | JPEG CAR PSNR / SSIM |
|---|---|---|---|---|
| DnCNN (Zhang et al., 2017) | 83.18 | 41.19 | 30.46 | - / - |
| DRUNet (Zhang et al., 2021) | 172.39 | 84.27 | 32.34 | 32.65 / 0.8919 |
| SwinIR (Liang et al., 2021) | 5689.86 | 2939.34 | 32.62 | 32.74 / 0.8926 |
| Restormer (Zamir et al., 2022) | 1201.53 | 623.15 | 32.71 | - / - |
| **ShareFormer (Ours)** | **806.97** | **424.70** | **32.79** | **32.75 / 0.8945** |

Table 10: Inference latency of different methods on different devices for image denoising and JPEG CAR tasks. The results are evaluated on Urban100 dataset.

| Method | Scale | RTX2080TI (ms) | RTX3090 (ms) | RTX4090 (ms) | Urban100 PSNR / SSIM |
|---|---|---|---|---|---|
| RCAN (Zhang et al., 2018b) | | 898.39 | 585.03 | 291.93 | 33.34 / 0.9384 |
| DRLN (Anwar & Barnes, 2020) | | 1317.69 | 731.98 | 443.34 | 33.37 / 0.9390 |
| HAN (Niu et al., 2020) | | 2645.60 | 1441.23 | 857.40 | 33.35 / 0.9385 |
| NLSA (Mei et al., 2021) | | N/A | 1345.97 | 949.69 | 33.42 / 0.9394 |
| SwinIR (Liang et al., 2021) | ×2 | 2245.83 | 1340.49 | 716.12 | 33.81 / 0.9427 |
| ELAN* (Zhang et al., 2022) | | 1824.84 | 1039.02 | 581.74 | 33.44 / 0.9391 |
| EDT (Li et al., 2023b) | | 3819.79 | 2439.41 | 1340.65 | 33.80 / 0.9425 |
| ART-S (Zhang et al., 2023) | | N/A | N/A | N/A | 34.02 / 0.9437 |
| **ShareFormer (Ours)** | | **1744.79** | **939.22** | **470.75** | **34.22 / 0.9451** |
| RCAN (Zhang et al., 2018b) | | 458.43 | 283.36 | 139.68 | 29.09 / 0.8702 |
| DRLN (Anwar & Barnes, 2020) | | 656.45 | 353.05 | 203.16 | 29.21 / 0.8722 |
| HAN (Niu et al., 2020) | | 1235.97 | 686.49 | 347.29 | 29.10 / 0.8705 |
| NLSA (Mei et al., 2021) | | 1268.14 | 657.33 | 395.45 | 29.25 / 0.8726 |
| SwinIR (Liang et al., 2021) | ×3 | 1002.33 | 605.45 | 327.77 | 29.75 / 0.8826 |
| ELAN* (Zhang et al., 2022) | | 824.09 | 456.57 | 275.87 | 29.32 / 0.8745 |
| EDT (Li et al., 2023b) | | 1623.71 | 993.25 | 570.21 | 29.72 / 0.8814 |
| ART-S (Zhang et al., 2023) | | N/A | 1682.21 | 853.57 | 29.86 / 0.8830 |
| **ShareFormer (Ours)** | | **775.63** | **422.62** | **218.79** | **29.95 / 0.8851** |
| RCAN (Zhang et al., 2018b) | | 238.36 | 178.01 | 84.07 | 26.82 / 0.8087 |
| DRLN (Anwar & Barnes, 2020) | | 342.05 | 202.62 | 115.17 | 26.98 / 0.8119 |
| HAN (Niu et al., 2020) | | 698.44 | 388.71 | 177.65 | 26.85 / 0.8094 |
| NLSA (Mei et al., 2021) | | 708.92 | 377.51 | 174.88 | 26.96 / 0.8109 |
| SwinIR (Liang et al., 2021) | ×4 | 493.10 | 293.25 | 140.05 | 27.45 / 0.8254 |
| ELAN* (Zhang et al., 2022) | | 432.24 | 236.95 | 130.45 | 27.13 / 0.8167 |
| EDT (Li et al., 2023b) | | 982.94 | 590.26 | 301.85 | 27.46 / 0.8246 |
| ART-S (Zhang et al., 2023) | | N/A | 547.52 | 285.60 | 27.54 / 0.8261 |
| **ShareFormer (Ours)** | | **400.20** | **211.10** | **113.99** | **27.64 / 0.8294** |

Table 11: Inference latency for classical image SR.

| Method | Scale | RTX2080TI (ms) | RTX3090 (ms) | RTX4090 (ms) | Urban100 PSNR / SSIM |
|---|---|---|---|---|---|
| IMDN (Hui et al., 2019) | | 47.86 | 39.71 | 20.27 | 32.17 / 0.9283 |
| RFDN-L (Liu et al., 2020a) | | 55.90 | 45.24 | 22.91 | 32.24 / 0.9290 |
| SwinIR (Liang et al., 2021) | ×2 | 487.61 | 310.3 | 222.78 | 32.76 / 0.9340 |
| ELAN (Zhang et al., 2022) | | 311.18 | 192.4 | 134.23 | 32.76 / 0.9340 |
| DLGSA-L (Li et al., 2023c) | | 507.81 | 224.2 | 165.98 | 32.94 / 0.9355 |
| **ShareFormer (Ours)** | | **251.69** | **149.6** | **111.09** | **33.13 / 0.9373** |
| IMDN (Hui et al., 2019) | | 28.05 | 15.34 | 10.27 | 28.17 / 0.8519 |
| RFDN-L (Liu et al., 2020a) | | 29.67 | 17.17 | 12.86 | 28.32 / 0.8547 |
| SwinIR (Liang et al., 2021) | ×3 | 246.06 | 134.52 | 97.40 | 28.66 / 0.8624 |
| ELAN (Zhang et al., 2022) | | 151.72 | 84.94 | 50.75 | 28.69 / 0.8624 |
| DLGSA-L (Li et al., 2023c) | | 239.66 | 102.31 | 76.36 | 28.83 / 0.8653 |
| **ShareFormer (Ours)** | | **123.06** | **67.51** | **40.54** | **29.01 / 0.8692** |
| IMDN (Hui et al., 2019) | | 13.57 | 11.21 | 7.96 | 26.04 / 0.7838 |
| RFDN-L (Liu et al., 2020a) | | 15.94 | 12.60 | 8.21 | 26.20 / 0.7883 |
| SwinIR (Liang et al., 2021) | ×4 | 132.10 | 78.20 | 52.66 | 26.47 / 0.7980 |
| ELAN (Zhang et al., 2022) | | 75.66 | 50.69 | 25.94 | 26.54 / 0.7982 |
| DLGSA-L (Li et al., 2023c) | | 134.46 | 86.33 | 52.85 | 26.66 / 0.8033 |
| **ShareFormer (Ours)** | | **62.93** | **41.47** | **20.77** | **26.83 / 0.8080** |

Table 12: Inference latency for lightweight image SR.

## B TRAINING DETAILS

In this section, we will present detailed training hyper-parameters for the main experiments to ensure that this paper can be perfectly reproduced.

**Datasets.** For image SR, following previous works (Liang et al., 2021), we use DIV2K (Agustsson & Timofte, 2017) and Flickr2K (Lim et al., 2017) as training data, Set5 (Bevilacqua et al., 2012), Set14 (Zeyde et al., 2012), BSD100 (Martin et al., 2001a), Uran100 (Huang et al., 2015) and Manga109 (Matsui et al., 2017) as test data. For image denoising and image JPEG compression artifact reduction, following previous works (Liang et al., 2021; Zamir et al., 2022), we use DIV2K, Flickr2K, BSD500 (Arbelaez et al., 2010), and WED (Ma et al., 2016) as training data. We use (C)BSD68 (Martin et al., 2001b), Kodak24 (Franzen, 2013), McMaster (Zhang et al., 2011), and Urban100 as test data for image denoising, Classic5 (Foi et al., 2007), and LIVE1 (Sheikh et al., 2006) as test data for image JPEG CAR. We reflect pad the input image to a size where the side length is an integer multiple of 64 before the test and crop back to the original size after the test.

**Loss Function.** We utilize Smooth $L_1$ loss (Huber, 1992) for image SR and Charbonnier loss (Charbonnier et al., 1994) for image denoising and JPEG compression artifact reduction. These loss functions are smoother and differentiable at zero compared to the $L_1$ loss more commonly used in earlier methods.

$$\text{Smooth } L_1 \text{ Loss}(I_G, \hat{I}) = \begin{cases} 0.5 \, (I_G - \hat{I})^2 & \text{for } |I_G - \hat{I}| \leq \delta, \\ |I_G - \hat{I}| - 0.5 \, \delta, & \text{otherwise.} \end{cases}$$

$$\text{Charbonnier Loss}(I_G, \hat{I}) = \sqrt{||I_G - \hat{I}||^2 + \epsilon^2}$$

(7)

where $\delta = 0.025 * \text{upsample scale} + 0.05$, $\epsilon = 0.001$.

**Implementation details.** Following SwinIR (Liang et al., 2021), the number of attention blocks, residual groups, and channels are generally set to 6, 6, and 180. The striped window sizes are 8 and 32. For lightweight image SR, we decrease the number of residual blocks, residual groups, and channels to 5, 4, and 60, respectively. We also decrease the window sizes to 8 and 16 in this case. The number of attention heads is 1 for all SPSA. Note that if there is relative position bias in the process of self-attention, it will also be shared in different layers, together with the attention maps. The details of the network structure for the image denoising and CAR tasks are exactly the same as for the Restormer (Zamir et al., 2022), except that the number of the SPSA's attention heads is 1.

**Training settings.** The AdamW optimizer with an initial learning rate $2 \times 10^{-4}$, batch-size 32, and iterations 500k trains the network from scratch for image SR. The initial learning rate is $5 \times 10^{-4}$ for

lightweight image SR, respectively. Following Restormer (Zamir et al., 2022), we use Progressive Learning, we start training with patch size $128 \times 128$, batch size 32, and learning rate $3 \times 10^{-4}$. We use horizontal and vertical flips as data augmentation and use cosine annealing as the learning strategy for all experiments.

| Task | Classical SR | Lightweight SR | Image Denoising & JPEG CAR |
|---|---|---|---|
| Batch-size | 8 | 64 | Progressive Learning(*) |
| Learning-rate | 2e-4 | 5e-4 | 3e-4 |
| Schedule | Cosine | Cosine | Cosine |
| Optim | AdamW (0.9, 0.999) | AdamW (0.9, 0.999) | AdamW (0.9, 0.999) |
| Training Iters | 500k | 500k | 300k |
| Warmup | None | None | None |
| Patch-size | 48 | 64 | Progressive Learning |
| Init | Default Initialization | Default Initialization | Default Initialization |
| augment | flip and rot 90 & 270 | flip and rot 90 & 270 | flip and rot 90 & 270 |
| grad clip | None | None | 0.01 |
| EMA | 0.999 | 0.999 | 0.999 |

Table 13: Training details of image SR, image denoising, and JPEG CAR. "Progressive Learning" means the progressive learning strategy used in Restormer. "Default Initialization" means PyTorch's default parameter initialization strategy.

## C    EXTRA RESULTS

In this section, we will provide additional details on the parameters and FLOPs of different methods used in the image denoising task, as well as present more examples of qualitative comparisons between image SR and denoising tasks.

### C.1    EXTRA QUANTITATIVE RESULTS

In this section, we present the parameters, FLOPs, latency, and PSNR metrics for various denoising methods, showcasing the greater efficacy of our approach.

| Method | Params | FLOPs | Latency | PSNR on image DN | PSNR on JPEG CAR |
|---|---|---|---|---|---|
| DnCNN | 0.56M | 0.55T | 83.18 | 31.24 | - |
| DRUNet | 32.64M | 2.15T | 172.39 | 31.69 | 34.41 |
| RNAN | 8.96M | N/A | N/A | - | - |
| SwinIR | 11.46M | 11.28T | 5689.86 | 31.78 | 34.52 |
| Restormer | 26.10M | 2.12T | 1201.53 | 31.79 | - |
| EDT-B | 11.63M | N/A | N/A | 31.76 | - |
| ShareFormer(ours) | 17.64M | 1.35T | 806.97 | 31.80 | 34.53 |

Table 14: The parameters, FLOPs, latency, and PSNR on CNSD68 $\sigma = 25$ image denoising (DN) task and Classic5 $q = 40$ image JPEG CAR task.

### C.2    EXTRA QUALITATIVE RESULTS

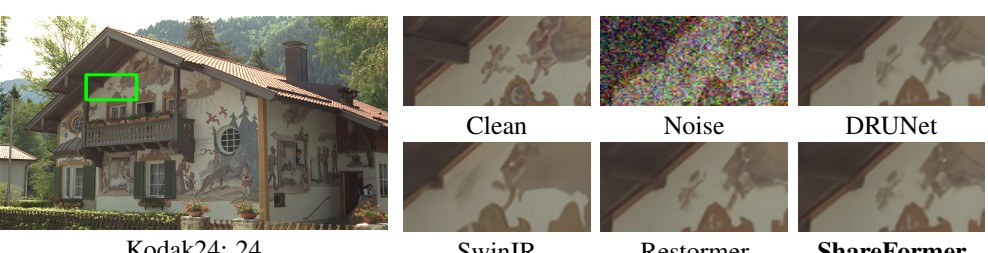

Figure 12: Qualitative comparison with recent SOTA methods on the $\sigma = 50$ *image Denoising* task.

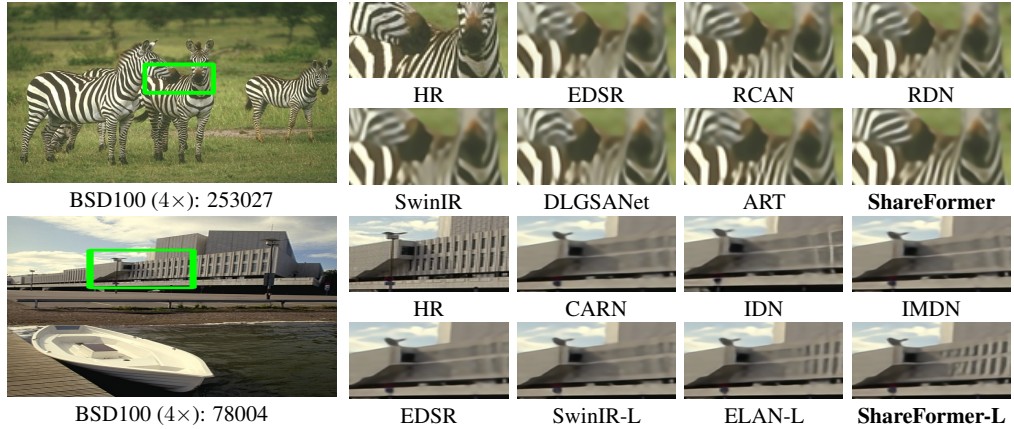

Figure 11: Qualitative comparison with recent SOTA methods on the $4\times$ *image SR* task. The top row is **classical image SR** results, and the bottom row is **lightweight image SR** results.

## D REDUNDANCY IN THE ATTENTION MAP

In this section, we present visualizations of the attention maps of SwinIR (Liang et al., 2021) pre-trained models on a $4\times$ lightweight image super-resolution task to demonstrate the significant redundancy in the Transformers for image restoration.

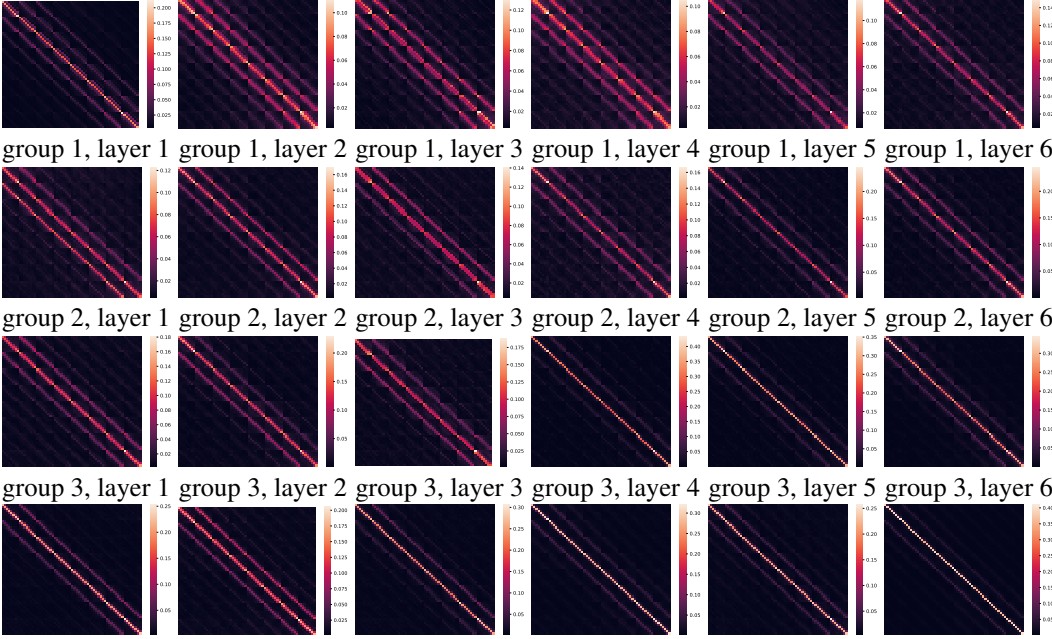

## E LESION STUDY

**Kendall Tau correlation coefficient.** We calculate the Kendall Tau correlation coefficient value of the weights of residual values' convolution layer.

**Effective Receptive Fields.** We calculated the ERF following RepLKNet (Ding et al., 2022). The ERF is defined to be $\partial f(x;\theta)/\partial x$, where $f(x;\theta)$ means the network and $\theta$ means the parameters of the network.

## F TRAINABILITY

In this section, we will analyze the trainability of ShareFormer in more detail.

**Quantitative comparison.** In the main text, we provide an adequate qualitative analysis of Share-Former's trainability. We use the NTK condition number as the quantitative result of networks' trainability. We calculated the NTK condition numbers for different networks in Tab. 15. ShareFormer has comparable trainability and higher performance with CNN-based methods, like RCAN (Zhang et al., 2018b).

| Method | NTK | Urban100 (PSNR / SSIM) |
|---|---|---|
| RCAN (Zhang et al., 2018b) | 107.03 | 26.82 / 0.8087 |
| SwinIR (Liang et al., 2021) | 1092.39 | 27.45 / 0.8254 |
| ELAN (Zhang et al., 2022) | 2446.48 | 27.13 / 0.8167 |
| DLGSANet (Li et al., 2023c) | 2025.87 | 27.17 / 0.8175 |
| **ShareFormer (Ours)** | **158.90** | **27.64 / 0.8294** |

Table 15: Trainability of ShareFormer and other methods on classical image SR $\times 4$ task.

We approximate the neural tangent kernel on the Set5 dataset by averaging over-block diagonal entries in the full NTK. Notice that the computation is based on the architecture at initialization without training.

