# OpenReview forum: "ShareFormer: Share Attention for Efficient Image Restoration"
_ICLR.cc/2024/Conference — Submitted to ICLR 2024_

### Official Review · Reviewer_DFQF · 2023-10-29

**Soundness:** 3 good
**Presentation:** 3 good
**Contribution:** 3 good
**Rating:** 6
**Confidence:** 4

**Summary:**

This paper tried to solve two problems: how to make Transformer faster and how to make Transformer's optimization faster.
For the first problem, they propose shared portion stripe attention (SPSA) to reduce the network latency up to 7x speedup.
For the second problem, they introduce residual connections to the value of SPSA.
In summary, they build a novel Transformer network: ShareFormer by SPSA with residual connections and gated united.

**Strengths:**

1. This paper reduced the computational complexity of stripe attention by Shared Portion Stripe Attention.

2. They proposed Residual Connections on Value, which offset the obstruction of the information flow throughout the network.

3. They reached similar or even better performance and fewer parameters compared with other methods.

**Weaknesses:**

1. In order to implement shared attention, they had to introduce residual connections on value and gated unit to control the extra complexity. I am concerned that the greater the complexity of the system, the higher the likelihood of training instability.

2. In Tables 4, 5, and 6, DRUNet reached similar performance and much lower latency compared with ShareFormer. To be honest, I prefer DRUNet in real applications with nearly 5x speedup, even though there is a 0.1~0.2 performance loss.

**Questions:**

1. Could you please also list the number of parameters of each model in Tables 4 and 6?

2. By adding V directly to the output, you're effectively giving more weight to the original values irrespective of the computed attention scores. This might dilute the effect of the attention mechanism, especially if the values in V dominate the weighted sum. Thus, what if introducing a trainable parameter to scale the residual connection instead of directly adding a residual connection?

---

> ### Author Response · Authors · 2023-11-19
> **Comments for your questions.**
>
> Thank you sincerely for your thorough review of our paper. We highly value the questions you raised. Below, we have provided detailed responses to each of them and made corresponding improvements to the paper. We hope our responses and revisions sufficiently address your concerns, and we genuinely appreciate your consideration for an improved rating of our paper.
>
> > For Q1, Could you please also list the number of parameters of each model in Tables 4 and 6?
>
> Certainly. We have included the parameter number and latency information for each model in the table below. The size of the model used for image denoising (corresponding to Table 4 in the main paper) and JPEG CAR (corresponding to Table 6 in the main paper) is the same. Therefore, we present these in a single table. Additionally, the details from this table have also been added to Appendix C.
>
> | Method            | Params | FLOPs  | Latency | PSNR on image denoising | PSNR on JPEG CAR |
> | ----------------- | ------ | ------ | ------- | ----------------------- | ---------------- |
> | DnCNN             | 0.56M  | 0.55T  | 83.18 ms  | 31.24                   | -                |
> | DRUNet            | 32.64M | 2.15T  | 172.39 ms | 31.69                   | 34.41            |
> | RNAN              | 8.96M  | N/A    | N/A     | -                       | -                |
> | SwinIR            | 11.46M | 11.28T | 5689.86 ms | 31.78                   | 34.52            |
> | Restormer         | 26.10M | 2.12T  | 1201.53 ms | 31.79                   | -                |
> | EDT-B             | 11.63M | N/A    | N/A     | 31.76                   | -                |
> | ShareFormer(ours) | 17.64M | 1.35T  | 806.97 ms | 31.80                   | 34.53            |
>
> PSNR on image denoising was calculated on the CBSD68 dataset, $\sigma=25$. PSNR on image JPEG CAR was calculated on the Classic5 dataset, $q=40$. "N/A" indicates that the corresponding models are too heavy for the NVIDIA RTX 3090 GPU. "-" means that the result is not available.
>
> > For Q2, Thus, what if introducing a trainable parameter to scale the residual connection instead of directly adding a residual connection?
>
> Thank you for your insightful suggestion. In fact, the introduction of a trainable residual connection is a design we had considered and experimented with earlier. However, we opted not to pursue this approach. The table below presents the experimental results on the classical $4 \times$ image super-resolution task, the test set is Urban100, where the residual scale was initialized to 1.
>
> | Methods                 | PSNR & SSIM on Urban100 |
> | ----------------------- | ---------- |
> | Vanilla residual        | 27.64 \| 0.8294 |
> | Learnable residual scale | 27.62 \| 0.8283 |
> | Learnable residual channel-wise | 27.68｜0.8293 |
>
> It is evident that incorporating diverse types of residuals does not result in a significant performance enhancement for ShareFormer in the classical $4 \times$ super-resolution task. This could be attributed to ShareFormer's intrinsic capacity to regulate value whereby supplementary trainable residual coefficients will not advance the network's performance (or a more detailed network initialization might be necessary).

---

> ### Author Response · Authors · 2023-11-19
> **Comments for your proposed weaknesses.**
>
> > For W1, I am concerned that the greater the complexity of the system, the higher the likelihood of training instability
>
> Thank you for raising an interesting research question. The correlation between model complexity and training stability is a topic of great importance. We have identified three initial perspectives on the issue:
> 1. Firstly, it is important to note that trainability is primarily intended to demonstrate that the model can converge rapidly[1], rather than stably[2]. In terms of stability of convergence, it may be more appropriate to use the sharpness as a metric [3].
> 2. In the three tasks discussed in this article, we did not notice any apparent variation in the training steadiness of models that differ in size. Nonetheless, we suspect that if the number of parameters is exceedingly high, the training instability will intensify as the intricacy of the model increases. Indeed, many studies relevant LLM scaling laws [4] have established that the more extensive the number of parameters, the less stable the training is.
> 3. It is a consensus within the deep learning field that more complex networks require more training tricks. there is an Figure 1 And Table 3&4 in RCAN-it[5] for complex RCANs within CNN networks, Figure 2 in ConvNeXT[6] also makes use of a number of tricks to give a pump in the performance of convolutional networks, and Table 8. in DeiT[7] proposes a series of tricks to make Transformer more stable to train. All these instances underscore the need for meticulously designed tricks in training to enhance the stability of complex networks.
>
> > For W2, In Tables 4, 5, and 6, DRUNet reached similar performance and much lower latency compared with ShareFormer.
>
> On this point of your concern, the choice between traditional CNNs and emerging Transformers is a widespread concern in the whole field. Here are a few of our perspectives:
>
> 1. As previously mentioned, DRUNet has approximately double the number of parameters compared to ShareFormer. This has led to a limitation in its practical application due to insufficient lightweight capabilities.
> 2. Unlike methods based on CNNs, models based on Transformers in the field of image restoration are still in the early and exploratory stages, with a considerable gap from practical application in industrial settings. Current efforts to improve inference speed and trainability, such as ELAN and our ShareFormer, represent steps toward achieving an optimal and practical Transformer network in the future. While it's just a small step, we believe our work holds significant relevance.
> 3. In reality, our method has made significant advances with progress of 0.38dB on the notoriously difficult Urban100 dataset, compared to DRUNet. We believe this improvement is significant and should be noteworthy in the selection of methods.
> 4. Unlike tasks like image classification and detection, numerical metrics are not the sole determinant of an image restoration method's effectiveness. Refer to Figure 10 in the main manuscript for a close-up view, which shows that our method excels in restoring the rich and intricate texture of the original image.
>
> ### References
>
> [1] Jacot A, Gabriel F, Hongler C. Neural tangent kernel: Convergence and generalization in neural networks[J]. Advances in neural information processing systems, 2018, 31.
>
> [2] Berlyand, Leonid and Jabin, Pierre-Emmanuel and Safsten, C. Alex "Stability for the training of deep neural networks and other classifiers" Mathematical Models and Methods in Applied Sciences 2021, 31, 11, 2345-2390.
>
> [3] Philip M. Long, Peter L. Bartlett. "Sharpness-Aware Minimization and the Edge of Stability". arXiv preprint arXiv:2309.12488.
>
> [4] Qin, Zhen and Li, Dong and Sun, Weigao and Sun, Weixuan and Shen, Xuyang and Han, Xiaodong and Wei, Yunshen and Lv, Baohong and Yuan, Fei and Luo, Xiao and others "Scaling transnormer to 175 billion parameters" arXiv preprint arXiv:2307.14995.
>
> [5] Lin, Zudi and Garg, Prateek and Banerjee, Atmadeep and Magid, Salma Abdel and Sun, Deqing and Zhang, Yulun and Van Gool, Luc and Wei, Donglai and Pfister, Hanspeter "Revisiting RCAN: Improved Training for Image Super-Resolution" arXiv preprint arXiv:2201.11279.
>
> [6] Zhuang Liu and Hanzi Mao and Chao-Yuan Wu and Christoph Feichtenhofer and Trevor Darrell and Saining Xie "A ConvNet for the 2020s" Proceedings of the IEEE/CVF Conference on Computer Vision and Pattern Recognition (CVPR) 2022.
>
> [7] Touvron, Hugo and Cord, Matthieu and Douze, Matthijs and Massa, Francisco and Sablayrolles, Alexandre and Jegou, Herve "Training data-efficient image transformers &amp; distillation through attention" International Conference on Machine Learning (ICML) 2021.

---

> ### Comment · Reviewer_DFQF · 2023-11-22
>
> Thanks for your response. The answer solved my concern about the comparison with DRUnet. It's glad to see you have already done the experiments of learnable residual scale which addressed my concern regarding the excessively high weights of value.
>
> I agree with reviewer nYLY's point W3, which is similar to my point W1: The ablation study is limited and fails to comprehensively assess the contribution of each component. Providing additional relevant ablation studies would make the argument more convincing

---

> > ### Author Response · Authors · 2023-11-22
> >
> > Thank you sincerely for your insightful feedback on our response. In response to your concern about the lack of ablation experiments, we have conducted thorough and comprehensive analyses. The detailed results have been presented in the attached table and are also included in Appendix C for your convenience. We believe these additional experiments address your queries and contribute positively to the overall quality of our paper.
> >
> > | Scale | Model       | Group Split Strategy | Share Attention | Residual on Value | CSAU | Params(M) | FLOPs(G) | Latency(ms) | PSNR   on Urban100 | SSIM on Urban100 |
> > | ----- | ----------- | -------------------- | --------------- | ----------------- | ---- | --------- | -------- | ----------- | ------------------ | ---------------- |
> > | x4    | SwinIR      |                      |                 |                   |      | 11.90     | 775.08   | 293.25      | 27.45              | 0.8254           |
> > | x4    | ShareFormer |                      |                 |                   |      | 10.09     | 625.37   | 290.82      | 27.43              | 0.8251           |
> > | x4    | ShareFormer | √                    |                 |                   |      | 10.09     | 625.37   | 299.33      | 27.55              | 0.8260           |
> > | x4    | ShareFormer | √                    | √               |                   |      | 8.91      | 553.71   | 247.33      | 27.41              | 0.8222           |
> > | x4    | ShareFormer | √                    | √               | √                 |      | 8.91      | 553.71   | 249.32      | 27.61              | 0.8280           |
> > | x4    | ShareFormer | √                    | √               |                   | √    | 7.67      | 478.46   | 210.87      | 27.43              | 0.8224           |
> > | x4    | ShareFormer | √                    | √               | √                 | √    | 7.67      | 478.46   | 211.10      | 27.64              | 0.8294           |
> >
> > We highly appreciate your time and effort invested in reviewing our work. Your feedback has undoubtedly enhanced the robustness of our research. We hope these revisions align with your expectations and contribute to an improved evaluation of our manuscript.

---

### Official Review · Reviewer_nYLY · 2023-10-30

**Soundness:** 3 good
**Presentation:** 3 good
**Contribution:** 2 fair
**Rating:** 5
**Confidence:** 5

**Summary:**

This paper proposes a new transformer architecture called ShareFormer for image restoration tasks like super-resolution and denoising. The key idea is to share attention maps between neighboring layers, which reduces computational cost and speeds up inference. Residual connections are added to preserve information flow. Experiments show ShareFormer achieves state-of-the-art accuracy with lower latency and better trainability than prior transformers.

**Strengths:**

•	The proposed ShareFormer delivers substantial improvements in efficiency, reducing latency by up to 2x compared to CNN models without compromising accuracy. This is achieved through an innovative technique of sharing attention maps between transformer layers to avoid redundant computations.
•	The method enhances trainability over other transformer architectures by introducing beneficial inductive biases, allowing faster convergence.
•	An additional strength is the generalizability of the approach, which is shown to be compatible with different attention mechanisms like shifted windows.

**Weaknesses:**

•	The performance exhibited by the proposed ShareFormer is indeed commendable, adeptly striking a balance between quality and speed. Nonetheless, I must express some reservations regarding the motivation behind the proposed backbone. To put it candidly, certain elements of the core module in ShareFormer appear reminiscent of concepts present in existing methodologies. For instance, the notions of Residual on V and the Reuse of Attention map are echoed in methods like Restormer and ELAN. Similarly, the group split strategy bears similarities to the one found in EfficientViT [1]. Thus, at a glance, ShareFormer seems to be a thoughtful amalgamation of pre-existing techniques. I would strongly recommend emphasizing the unique aspects and contributions of ShareFormer to underscore its originality within the broader landscape.
[1] Xinyu Liu, Houwen Peng, Ningxin Zheng, Yuqing Yang, Han Hu, Yixuan Yuan: EfficientViT: Memory Efficient Vision Transformer with Cascaded Group Attention. CVPR 2023: 14420-14430.
•	The gains in trainability from the residual connections could use more detailed analysis and intuition. The paper currently lacks insight into the underlying mechanisms enabling faster convergence.
•	The ablation study is limited and does not thoroughly evaluate the contribution of each component. More experiments could help tease apart the individual impact of techniques like SPSA and the gated units.
•	Important implementation details like dataset splits for training, validation, and testing are not provided. This makes reproducibility difficult.
•	Besides the Image Super-Resolution, the overall improvements appear relatively incremental over strong prior work like SwinIR and Restormer. The advances are not radically transformative.

**Questions:**

Some concerns are raised in Weakness.

---

> ### Author Response · Authors · 2023-11-19
> **Comment for your proposed weakness 1.**
>
> Thank you for dedicating time and effort to review our paper. Below, we address each of your points and have made corresponding adjustments to the manuscript based on your valuable suggestions. If our responses sufficiently address your queries, would you kindly consider enhancing the rating for our paper?
>
> > For W1, ShareFormer seems to be a thoughtful amalgamation of pre-existing techniques.
>
> Thank you for your commendation on the balance achieved between quality and speed in our proposed method. Regarding the innovation of the method, we would like to provide further details on a few points.
>
> 1. About Residual on V: I would like to clarify that ELAN and Restormer did not propose this innovation. There may be some misunderstanding. Further, we will demonstrate that residual connections on the feature value facilitate the shared attention operation to counteract its inherent drawbacks, while also ensuring higher performance and no impact on inference speed.  Experiments about $4 \times$ classical image super resolution were conducted with different numbers of shared layers, exploring two scenarios of residual connection on Value and other configurations. The results revealed the following findings.
>
> | Shared Layer | With residual v | Without residual value |
> | ------------ | --------------- | ---------------------- |
> | 2            | 27.64 \| 0.8294 | 27.43 \| 0.8224        |
> | 4            | 27.48 \| 0.8241 | 27.25 \| 0.8199        |
> | 6            | 27.45 \| 0.8234 | 27.20 \| 0.8181        |
>
> As you can see, if the attention maps are only shared between nearby layers, the model's performance will significantly decrease. This demonstrates why it's important and effective to have a residual connection on value.
>
> 2. About the group split strategy: the feature splitting approach in this paper differs from EfficientViT, which we will illustrate with a piece of PyTorch-style pseudo-code.
>
> Following the EfficientViT's [official code](https://github.com/microsoft/Cream/blob/main/EfficientViT/classification/model/efficientvit.py#L165-L179), they add the previous group's output to the next group's input. It also performs an additional grouping convolution on the query. The group split strategy of EfficientViT is:
>
> ```python
> for i, qkv in enumerate(self.qkvs):
>     if i > 0: # add the previous output to the input
>         feat = feat + feats_in[i]
>     feat = qkv(feat)
>     q, k, v = qkv.chunk(3, dim=1)
>     q = self.dws[i](q)
>     attn = (q.transpose(-2, -1) @ k) * self.scale
>     attn = attn.softmax(dim=-1)
>     feat = v @ attn.transpose(-2, -1)
>     feats_out.append(feat)
> x = self.proj(torch.cat(feats_out, 1))
> ```
>
> However, we refrained from adding the output of the former group to the latter, as we believe it is an ineffective (or detrimental) operation for the image restoration task due to the following reason. The feature groups will be inputted into window attentions of different window-sizes. The first group of features will be inputted into a window of size [32, 8] by default, while the second group of features will be inputted into a window of size [8, 32]. The windows will analyse and process texture information in different directions. According to Figure 4 in GRL [3], we could see that "image features in natural images are anisotropic." Therefore, two sets of features will be utilized for extracting heterogeneous information. Overlapping these features through addition will restrict the second set of window attention from learning useful knowledge, as its input includes texture information in the opposite direction. Here it's the group split strategy of ShareFormer:
>
> ```python
> qkvs = self.qkvs(feat).chunk(split_num, dim=1)
> feats_out = []
> for qkv in qkvs:
>     q, k, v = qkv.chunk(3, dim=1)
>     attn = (q.transpose(-2, -1) @ k) * self.scale
>     attn = attn.softmax(dim=-1)
>     feat = v @ attn.transpose(-2, -1)
>     feats_out.append(feat)
> x = self.proj(torch.cat(feats_out, 1))
> ```
>
> We used EfficientViT's group split strategy to train ShareFormer on the classical $4 \times$ image super-resolution task, and the experimental results proved our point.
>
> | Group Split Strategy | PSNR on Urban100 | SSIM on Urban100 |
> | -------------------- | ---------------- | ---------------- |
> | EfficientViT         | 27.48            | 0.8250           |
> | ShareFormer (ours)   | 27.64            | 0.8294           |
>
> [1] Yang G. Tensor programs ii: Neural tangent kernel for any architecture[J]. arXiv preprint arXiv:2006.14548.
>
> [2] Yang G, Littwin E. Tensor programs iib: Architectural universality of neural tangent kernel training dynamics[C]//International Conference on Machine Learning. PMLR, 2021: 11762-11772.
>
> [3] Y. Li, et al., "Efficient and Explicit Modelling of Image Hierarchies for Image Restoration," in 2023 IEEE/CVF Conference on Computer Vision and Pattern Recognition (CVPR), Vancouver, BC, Canada, 2023 pp. 18278-18289.

---

> > ### Author Response · Authors · 2023-11-22
> > **Comment for ablation studies.**
> >
> > Therefore, based on the thorough and comprehensive analyses of ablation, we have compiled the following result table, which will also be supplemented in Appendix C.
> >
> >
> > | Scale | Model       | Group Split Strategy | Share Attention | Residual on Value | CSAU | Params(M) | FLOPs(G) | Latency(ms) | PSNR   on Urban100 | SSIM on Urban100 |
> > | ----- | ----------- | -------------------- | --------------- | ----------------- | ---- | --------- | -------- | ----------- | ------------------ | ---------------- |
> > | x4    | SwinIR      |                      |                 |                   |      | 11.90     | 775.08   | 293.25      | 27.45              | 0.8254           |
> > | x4    | ShareFormer |                      |                 |                   |      | 10.09     | 625.37   | 290.82      | 27.43              | 0.8251           |
> > | x4    | ShareFormer | √                    |                 |                   |      | 10.09     | 625.37   | 299.33      | 27.55              | 0.8260           |
> > | x4    | ShareFormer | √                    | √               |                   |      | 8.91      | 553.71   | 247.33      | 27.41              | 0.8222           |
> > | x4    | ShareFormer | √                    | √               | √                 |      | 8.91      | 553.71   | 249.32      | 27.61              | 0.8280           |
> > | x4    | ShareFormer | √                    | √               |                   | √    | 7.67      | 478.46   | 210.87      | 27.43              | 0.8224           |
> > | x4    | ShareFormer | √                    | √               | √                 | √    | 7.67      | 478.46   | 211.10      | 27.64              | 0.8294           |

---

> > ### Comment · Reviewer_nYLY · 2023-11-23
> >
> > The authors are attempting to use code differences to illustrate the differences between the proposed method and previous methods, which means that the major idea is essentially consistent with the previous methods, with only minor differences in implementation details. In this case, such innovation is insignificant. I noticed that one reviewer mentioned that shared attention map mechanisms have already been proposed in ELAN, but the authors do not seem to have provided any response.

---

> > > ### Author Response · Authors · 2023-11-23
> > >
> > > Maybe I misunderstood your valuable opinion. We further give the following views.
> > >
> > > 1. You mention the importance of numerical metrics. I agree that numerical metrics are certainly one of the easiest ways to compare the merits of methods, but please note that our method is designed to be more efficient, not higher on PSNR or SSIM without seeing qualitative results. If numerical metrics about performance are the only metrics used to evaluate whether a paper is accepted, should all papers that do not achieve a classification accuracy of 90 on the ImageNet1k dataset be rejected? So, should all articles whose IoU metric does not reach 66 on the COCO dataset be rejected? Should all articles doing LLM be rejected because they do not perform as well as ChatGPT4? At the same time, I have observed that you have intentionally ignored our contribution to efficiency. Do you think Parameters, FLOPs, and Latency should not be used as an evaluation metric? We are eagerly looking forward to your views on this issue.
> > > 2. We have explained in detail the motivation and reasons for our use of feature grouping, given that the method you mentioned is not actually suitable for the image restoration task. When module A is not suitable for a certain task, a simple change can prevent this bad thing from happening. This change cannot be said to be meaningless.
> > > 3. Regarding your accusation that the article is not novel enough. Our paper spends an entire section on the effect of residual connections on value, and shared attention is only one component of our work, not our main innovation. If you completely reject the value of the other modules of the paper because we used share attention, does that mean you think all papers using residual connections should be rejected because almost all networks use residual connections and they are not novel enough?
> > >
> > > In summary, we sincerely hope that the potential contributions of our work to the overall research community will be more fully considered. We believe that you can provide a more rational and objective evaluation. We again thank you for taking the time and effort to review our manuscript. Thank you!

---

> ### Author Response · Authors · 2023-11-19
> **Comments for your proposed weaknesses 2-4.**
>
> > For W2, The gains in trainability from the residual connections could use more detailed analysis and intuition.
>
> Sharing Attention reduces redundancy while also reducing the efficiency of information flow by restricting the channels for layer-by-layer information flow (you can think of the operation `Share` as a residual connection without residual branches, i.e., $y = f(x) + x \ \text{and} \ f(x) = 0$). In order to improve the efficiency of information flow while controlling the amount of computation, we introduce residual on V.
>
> However, due to space constraints, we are only able to explain the trainability of residual on V from an experimental point of view rather than a mathematical one. The NTK spectral analysis[1,2] of the Transformer with shared attention and residual on V is exactly where we will direct our future work.
>
> > For W3, The ablation study is limited and does not thoroughly evaluate the contribution of each component.
>
> We add ablation experiments on whether CSAU is more effective, please see the table below:
>
> |                       | Param | Latency | PSNR on Urban100 SR x4   |
> | --------------------- | ----- | ------- | --------------- |
> | Shared Attention+MLP  | 9.43M | 238.47 ms | 27.59 \| 0.8275 |
> | Shared Attention+GDFN | 8.91M | 249.32 ms | 27.61 \| 0.8280 |
> | CSAU(ours)            | 7.67M | 211.10 ms | 27.64 \| 0.8294 |
>
> It is evident that CSAU is more efficient when fewer parameters are used. In fact, CSAU serves as an efficient approach to reducing excessive computation in the Feed Forward Network (FFN) section. It accomplishes this by relocating the gating operation before the FFN computation process, which streamlines the FFN into an identity module, resulting in faster execution.
>
> > For W4, Important implementation details like dataset splits for training, validation, and testing are not provided. This makes reproducibility difficult.
>
> The specific training datasets, training and testing details have been provided in Appendix B. Here, we present the hyperparameters for training for the four tasks outlined in the article. This is intended to facilitate the reproduction of this work by others and to address any questions that may arise.
>
> | Task           | Classical Super Resolution        | Lightweight Super Resolution      | Image Denoising & JPEG CAR        |
> | -------------- | --------------------------------- | --------------------------------- | --------------------------------- |
> | Batch-size     | 8                                 | 64                                | Progressive Learning in Restormer |
> | Learning-rate  | 2e-4                              | 5e-4                              | 3e-4                              |
> | Schedule       | Cosine                               | Cosine                               | Cosine                               |
> | Optim          | AdamW (0.9, 0.999)                | AdamW (0.9, 0.999)                | AdamW (0.9, 0.999)                |
> | Training Iters | 500k                              | 500k                              | 300k                              |
> | Warmup         | None                              | None                              | None                              |
> | Patch-size     | 48                                | 64                                | Progressive Learning in Restormer |
> | Init           | Default Initialization of PyTorch | Default Initialization of PyTorch | Default Initialization of PyTorch |
> | augment        | flip and rot 90 & 270             | flip and rot 90 & 270             | flip and rot 90 & 270             |
> | grad clip      | None                              | None                              | 0.01                              |
> | EMA            | 0.999                             | 0.999                             | 0.999                             |
>
> [1] Yang G. Tensor programs ii: Neural tangent kernel for any architecture[J]. arXiv preprint arXiv:2006.14548.
>
> [2] Yang G, Littwin E. Tensor programs iib: Architectural universality of neural tangent kernel training dynamics[C]//International Conference on Machine Learning. PMLR, 2021: 11762-11772.

---

> ### Author Response · Authors · 2023-11-19
> **Comment for your proposed weakness 5.**
>
> > For W5, Besides the Image Super-Resolution, the overall improvements appear relatively incremental over strong prior work like SwinIR and Restormer.
>
> Your concern pertains to a wider challenge within the field of image restoration. Let me elaborate on a couple of aspects:
>
> 1. Image restoration poses unique challenges compared to other high-level visual tasks in terms of defining metrics and achieving improvements. As stated in the discussion of this survey [4]: "More accurate metrics need to be found for image denoising. PSNR and SSIM are popular metrics for the task of image restoration. PSNR suffers from excessive smoothing, which is very difficult to recognize indistinguishable images. SSIM depends on brightness, contrast and structure, and therefore cannot accurately evaluate image perceptual quality." Metrics such as PSNR are not as absolute as accuracy in classification tasks or IoU in detection tasks.
> 2. The assessment of image quality remains challenging due to the lack of a flawless objective metric. Conventional measures, such as PSNR and SSIM, oversimplify the multidimensional evaluation of image quality. Therefore, the numerical differences among cutting-edge techniques, especially those founded on Transformers, become less significant. However, Fig. 10 in our paper and detailed perceptual comparisons in Appendix C reveal our approach's ability to recover finer details, including dense dots in real images. This demonstrates the visual superiority of our method over those of the competition.
> 3. It is crucial to emphasize that we aim to achieve an optimal Pareto balance between performance and latency rather than focusing solely on performance gains. In this respect, our method has a profound impact on the efficient use of algorithms, notably extensive Transformer-based models, in authentic industrial environments.
>
> [4] Chunwei Tian, Lunke Fei, Wenxian Zheng, Yong Xu, Wangmeng Zuo, Chia-Wen Lin, Deep learning on image denoising: An overview, Neural Networks, Volume 131, 2020, Pages 251-275, ISSN 0893-6080.

---

> > ### Comment · Reviewer_nYLY · 2023-11-23
> >
> > Anyway, PSNR and SSIM are still very good evaluation metrics. A good and effective method generally performs well in both of these metrics. If there is no significant improvement in these two metrics, it is actually difficult to convince me. Additionally, your paper does not demonstrate the advantages of the proposed method in other metrics.

---

### Official Review · Reviewer_F9tc · 2023-10-31

**Soundness:** 2 fair
**Presentation:** 2 fair
**Contribution:** 2 fair
**Rating:** 5
**Confidence:** 5

**Summary:**

This paper mainly proposes Shared Portion Stripe Attention (SPSA), which shares attention map in $(l-1)$-th and $l$-th layers and residually connects $\textit{value}$ of attention to intermediate attention output. To show the importance of the proposed residually connected $\textit{value}$, a Lesion study has been conducted. The authors also propose Combine SPSA with Gated Unit (CSGU) to enhance the existing GDFN module. ShareFormer shows comparable or improved performance when compared to state-of-the-art image restoration methods.

**Strengths:**

[S1] The references and related works cited by the authors are very recent and relevant.

[S2] The Lesion study presented in Sec.4.1 is comprehensive for demonstrating the proposed shared attention is highly related to ensemble behavior of the sequentially placed attention mechanism. Deleting or permutating some self-attention layers could smoothly increase the reconstruction loss. This can be the evidence that the shared attention is not the ensembles of shallow networks.

[S3] The performances of ShareFormer for large SR and lightweight SR are notably enhanced, compared to recent SOTA methods. These results were achieved with smaller model size, fewer computations, and faster speed than the others. (But an unfair issue remain. See question Q4.)

**Weaknesses:**

[W1] The presentation of text and figures in this paper should become clearer and more understandable. See Q1, Q2, and Q3.

[W2] The paper lacks some ablation studies proving the importance of the proposed components, such as CSAU. The authors should study architecture variants if they hope to clarify that CSAU comes not from insufficient considerations but from careful construction. And the efficiency and effectiveness differences between the original GDFN and the enhanced CSAU must be provided.

[W3] Most importantly, the core parts of ShareFormer, shared attention, seem not novel. In ELAN, one of the sota methods, shared attention map mechanisms have been already proposed. Moreover, it has been already shown how sharing attention maps in more than one layers can impact on the performance and efficiency of the model (related to your Tab.8).

[W4] A potential unfair issue is observed with respect to SR. See Q4.

[W5] Despite the improved inference speed, the performance gains of grayscale denoising, color denoising, and JPEG CAR are not significant.

**Questions:**

[Q1] Where is the exact part that the shared attention is operated? Eq.(4), (5) apparently reveals that this operates in $(l-1)$-th and $l$-th SPSA blocks. However, Fig.2 illustrates attention map of SPSA is shared to CSAU, while Fig.3 depicts sharing attention map appears after residual connection on $\textit{value}$. The reviewer thinks that the explanation of Eq.(4), (5) is the correct case the authors intended, while the phrase, "**sharing attention map**", in Fig.2 and Fig.3 is confusing. If it’s right, “sharing” can be omitted. Or not, please let me know what your first intention with respect to the exact parts of sharing attention map is.

[Q2] This question is related to Sec.4.2. From ConViT, how can you draw the conclusion that the concentrated ERF implies an amplified locality bias of the network? The reviewer thinks that it is not sufficient for the authors to claim that this fact shows the locality bias of the residually connected $\textit{value}$ in ShareFormer. I cannot find the acceptable evidence from ConViT paper. Don’t you have any other evidence justifying your claim, such as visualization?

[Q3] In Appendix D, why did you compare the attention maps of layers 1 and 3? ShareFormer shares attention map in layer-order-number pairs (0, 1), (2, 3), …, (2n-2, 2n-1), respectively. Thus, comparison of attention maps in “layers 0 and 1” or “layers 2 and 3” is more compelling to demonstrate attention map redundancy. Additionally, I recommend the authors to compare attention redundancies of the cases where the shared attention is “employed” and “not employed” in the proposed ShareFormer, instead of SwinIR cases.

[Q4] Did you apply the Progressive Learning to large and lightweight SR tasks with training patch size of from 128 to 384? However, the comparative methods, SwinIR, ELAN, and DLGSA, used smaller patch size, such as 48x48 and 64x64. I am concerned that this leads to a potential unfairness issue.



**Minor issue (not necessary to be mentioned in author rebuttal, if the authors struggle to a limit on the number of characters of rebuttal.)**

(1) Fig.2 omits (a) and (b) mark, while the caption uses (a) and (b).

(2) In the last to second sentence of Tab.1 caption, Restormer never used window attention. Moreover, you should mention what is MHDA, which seems a typo. (In Restormer, MDTA was used, and MHDA was not shown in SwinIR.)

---

> ### Author Response · Authors · 2023-11-19
> **Comments for your Q1, Q2, and Q3.**
>
> Thank you for your meticulous review and valuable insights into our paper. Below, we will respond to each of the questions you raised and make corresponding modifications to the manuscript. If our responses address your concerns and the revised manuscript meets your satisfaction, we kindly ask for your consideration in potentially enhancing the rating of our paper. Your feedback is immensely appreciated.
>
>
> > For Q1: Where is the exact part that the shared attention is operated?
>
> We have made appropriate changes to Figures 2 and Figure 3 and their captions in the article to make them clearer.
>
> In fact, both Attention Map and Feature Value obtained in SPSA are fed into CSAU. For easier understanding, we summarise the whole procedure as Share Attention Block (SAB).In SAB, there exist two layers, the first one contains two modules: SPSA, GDFN, and the second one contains one module, which is CSAU. Among them, SPSA is used to compute the Attention Map $Attn\_{l-1}$ and Feature $V\_{l-1}$, while GDFN is classically used as Feed Forward Network (FFN); for the $l$th layer, which is the CSAU module, it receives the output of GDFN $X\_{l}$, SPSA's attention map $Attn\_{l-1}$ and feature $V\_{l-1}$, and obtains the output according to Eq. 5 and Eq. 7 in the article.
>
> Specifically, we can use the following formulas for further explanation:
>
> $$
> Q_{l-1}, K_{l-1} = W_q(X_{l-1}), W_k(X_{l-1});
> $$
> $$
> Attn_{l-1} = Softmax((Q_{l-1}K_{l-1}^T)/s);
> $$
> $$
> V_{l-1} = W_v(X_l{l-1});
> $$
> $$
> X_{l} = W_{l-1}(Attn_{l-1} V_{l-1}) + X_{l-1};
> $$
>
> The above four lines of equations represent the specific work of the SPSA;
>
> $$
> X_{l} = GDFN(X_{l});
> $$
>
> The formula above represents the specific work of the GDFN;
>
> $$
> G_{l}, V_{l} = W_g X_{l}, W_v{X_{l}} + V_{l-1};
> $$
> $$
> Attn_l = Attn_{l-1};
> $$
> $$
> X_{l+1} = W_{l}(G_{l} \odot (Attn_l V_l)) + X_l;
> $$
>
> The above three lines of equations represent the specific work of the CSAU.
>
> We hope this explanation gives you a clearer understanding of our network structure, and you can also view the updated Figure 2 and Figure 3 for a more intuitive understanding of the model's design options.
>
> > For Q2: From ConViT, how can you draw the conclusion that the concentrated ERF implies an amplified locality bias of the network?
>
> We suspect there might be a misunderstanding in our presentation. In Section 4.2 of our paper, the statement is as follows: "As elucidated by Barzilai et al. (2023), the ensemble behavior of the network results in its ERF carrying more weight towards the central region of the receptive field." Therefore, we did not draw this conclusion from ConViT. This particular conclusion was obtained from Section 5.2 of "A kernel perspective of skip connections in convolutional networks" (Barzilai et al. 2023) [[1]](https://openreview.net/forum?id=6H_uOfcwiVh), a result that has been rigorously proven mathematically.
>
> > For Q3, In Appendix D, why did you compare the attention maps of layers 1 and 3?
>
> Thank you for your inquiry, which brings up a crucial point.
>
> 1. In Appendix D, we compared layers 1 and 3 because these two layers happened to exhibit an apparent redundancy, making it more beneficial to illustrate for readers' comprehension. However, we did not anticipate that this would lead to misunderstandings, and we sincerely apologize for that. To enhance clarity and aid understanding, we have now provided a complete set of attention maps visualization for all 24 layers of SwinIR-L in Appendix D. In reality, redundancy in attention computation might occur between adjacent layers (as shown in layer 4 and layer 5 of block 3 in the updated Appendix D) or even across distant layers (such as the previously compared layer 1 and layer 3 of block 1, and the layer 0 and layer 5 of block 4 in the updated Appendix D). The reasons for redundancy appearing across distant layers and how to determine which two layers exhibit significant redundancy are valuable questions that we acknowledge and plan to address in our future works.
> 2. The use of SwinIR for comparison was not intended to convey any specific meaning. We simply aimed to illustrate that redundancy in attention computation is prevalent in various Transformers, and SwinIR, being one of the most famous and representative Transformer networks in the field of image restoration, served as a natural example to demonstrate this phenomenon. However, we also think your suggestion is meaningful. We are currently training ShareFormer without shared attention maps and observing the computational redundancy it introduces. We plan to update these results in the Appendices within the next two days.
>
> [1] Daniel Barzilai, Amnon Geifman, Meirav Galun, Ronen Basri. "A Kernel Perspective of Skip Connections in Convolutional Networks" ICLR 2023.

---

> > ### Author Response · Authors · 2023-11-19
> > **Comments for your Q4.**
> >
> > > For Q4, Did you apply the Progressive Learning to large and lightweight SR tasks with training patch size of from 128 to 384?
> >
> > Your concern about unfairness in our comparison is entirely inexistent. First of all, we apologize for not providing specific training settings in the paper. In the initial draft, constrained by space, we only provided general descriptions like "Following SwinIR" and "exactly the same as for the Restormer" in Appendix B. Regarding your raised points:
> >
> > 1. We did not use Progressive Learning for either classical or lightweight SR. Progressive Learning was only used for image denoising and JPEG CAR.
> > 2. For training on classical SR, the patch size was 48 \* 48, and for lightweight SR, the patch size was 64 \* 64.
> > 3. Our training code is fully consistent with the open-source repository [BasicSR](https://github.com/XPixelGroup/BasicSR)(for SR) and [Restormer](https://github.com/swz30/Restormer)(for Denoising & JPEG CAR).
> >
> > Additionally, we are providing the specific settings and hyperparameters used for training on different tasks below, aiming to comprehensively address your concerns. This table has also been added to Appendix B to enhance the fairness and reproducibility of our method.
> >
> > | Task           | Classical Super Resolution        | Lightweight Super Resolution      | Image Denoising & JPEG CAR        |
> > | -------------- | --------------------------------- | --------------------------------- | --------------------------------- |
> > | Batch-size     | 8                                 | 64                                | Progressive Learning in Restormer |
> > | Learning-rate  | 2e-4                              | 5e-4                              | 3e-4                              |
> > | Schedule       | Cosine                               | Cosine                               | Cosine                               |
> > | Optim          | AdamW (0.9, 0.999)                | AdamW (0.9, 0.999)                | AdamW (0.9, 0.999)                |
> > | Training Iters | 500k                              | 500k                              | 300k                              |
> > | Warmup         | None                              | None                              | None                              |
> > | Patch-size     | 48                                | 64                                | Progressive Learning in Restormer |
> > | Init           | Default Initialization of PyTorch | Default Initialization of PyTorch | Default Initialization of PyTorch |
> > | augment        | flip and rot 90 & 270             | flip and rot 90 & 270             | flip and rot 90 & 270             |
> > | grad clip      | None                              | None                              | 0.01                              |
> > | EMA            | 0.999                             | 0.999                             | 0.999                             |

---

> ### Author Response · Authors · 2023-11-19
> **Comments for your proposed Weaknesses**
>
> > For W1, The presentation of text and figures in this paper should become clearer and more understandable. See Q1, Q2, and Q3.
>
> I hope the above answers on Q1-Q3 have allayed your concerns.
>
> > For W2, The paper lacks some ablation studies proving the importance of the proposed components, such as CSAU.
>
> Thanks to your kind reminder, we supplemented the ablation experiment with this module of CSAU on classical image $4 \times$ super-resolution task. The test dataset is Urban100. The results are as follows:
>
> |                       | Param | Latency | PSNR & SSIM     |
> | --------------------- | ----- | ------- | --------------- |
> | Shared Attention+MLP  | 9.43M | 238.47 ms  | 27.59 \| 0.8275 |
> | Shared Attention+GDFN | 8.91M | 249.32 ms | 27.61 \| 0.8280 |
> | CSAU(ours)            | 7.67M | 211.10 ms | 27.64 \| 0.8294 |
>
> It's clear that CSAU works better and faster with fewer parameters. In fact, CSAU is an efficient way to avoid excessive computation in the Feed Forward Network (FFN) part, because it moves the gating operation from the middle of the FFN computation process to before the FFN computation process and simplifies the FFN into an identity module.
>
> > For W3, shared attention seems not novel.
>
> The core of our approach is to implement residual connections on the Value element while completely sharing the attention feature map of the network among adjacent layers. This allows the shared attention operation to offset the shortcomings brought about by it, all the while ensuring that there isn't excessive loss of accuracy and running speed remains unaffected.  We conducted experiments on varying numbers of shared layers for the two scenarios of residual connection on Value or otherwise in classical image $4 \times$ super-resolution task, and the findings are as follows (the test set is the Urban100):
>
> | Shared Layer | With residual value | Without residual value |
> | ------------ | --------------- | ---------------------- |
> | 2            | 27.64 \| 0.8294 | 27.43 \| 0.8224        |
> | 4            | 27.48 \| 0.8241 | 27.25 \| 0.8199        |
> | 6            | 27.45 \| 0.8234 | 27.20 \| 0.8181        |
>
> As you can see, if the attention maps are only shared between nearby layers, the model's performance will significantly decrease. This demonstrates why it's important and effective to have a residual connection on value.
>
> > For W4, A potential unfair issue is observed with respect to SR. See Q4.
>
> I hope the above answer on Q4 has solved your problem.
>
> > For W5, Despite the improved inference speed, the performance gains of grayscale denoising, color denoising, and JPEG CAR are not significant.
>
> Your concern addresses a broader challenge inherent to the entire domain of image restoration. Allow me to shed light on a few aspects:
>
> 1. Image restoration, distinct from other high-level visual tasks, faces unique challenges in defining metrics and achieving improvements. As stated in the discussion of this review [1]: "More accurate metrics need to be found for image denoising. PSNR and SSIM are popular metrics for the task of image restoration. PSNR suffers from excessive smoothing, which is very difficult to recognize indistinguishable images. SSIM depends on brightness, contrast and structure, and therefore cannot accurately evaluate image perceptual quality." Metrics such as PSNR are not as absolute as accuracy in classification tasks or IoU in detection tasks.
> 2. The evaluation of image quality remains a complex task, given the absence of a perfect objective metric. Traditional measures like PSNR and SSIM oversimplify the multidimensional assessment of image quality. Consequently, the numeric variations among state-of-the-art methods, particularly those based on Transformers, become less prominent. However, as illustrated in Fig. 10 of our paper and detailed perceptual comparisons in Appendix C, our approach excels in recovering finer details, such as dense dots in real images. This showcases the visual superiority of our method over others.
> 3. Importantly, our emphasis lies in striking an optimal Pareto balance between performance and latency, rather than pursuing absolute performance gains. In this regard, our approach holds significant implications for the effective application of algorithms, especially large Transformer-based models, in real-life industrial settings.
>
> [1] Chunwei Tian, Lunke Fei, Wenxian Zheng, Yong Xu, Wangmeng Zuo, Chia-Wen Lin, Deep learning on image denoising: An overview, Neural Networks, Volume 131, 2020, Pages 251-275, ISSN 0893-6080.

---

> ### Author Response · Authors · 2023-11-19
> **Comments for your proposed minor issues.**
>
> > Fig.2 omits (a) and (b) mark, while the caption uses (a) and (b).
>
> Thank you sincerely for your careful review of our manuscript. We have addressed the identified issue in the paper accordingly.
>
> > Restormer never used window attention.
>
> Thank you for pointing out this issue. It was, in fact, a typo and a description that had been deprecated in the early iterations of the manuscript. It was not relevant to the content discussed in the paper. We have now removed it from the manuscript.

---

> > ### Comment · Reviewer_F9tc · 2023-11-22
> > **comments for the author rebuttal**
> >
> > [Q1] The revised Figs. 2 and 3 enhance the clarity of the proposed mechanism compared to the previous version, addressing my initial concern.
> >
> > [Q2] My concern has been fully addressed and resolved. However, there is a potential for misunderstanding due to the placement of the reference of sentence in Sec. 4.2: "This observation implies anamplified locality bias of the network, which has been proven to be capable of facilitating the training(d’Ascolietal.,2021).". It appears that the conclusion is drawn from ConViT. Therefore, I recommend a revision for clarity.
> >
> > [Q3] The revised figure in Appendix D compares attention maps in SwinIR. However, the redundancy of attention maps in adjacent layers does not seem apparent. While some are similar, there are actual differences when considering the scales of color bars. I believe the revised figure falls short of explaining the main problem highlighted by this paper.
> >
> > [Q4] My concern has been fully addressed and resolved.
> >
> > [W1] See the comments on Q1,Q2,Q3.
> >
> > [W2] The supplemented ablation study on the proposed CSAU does not sufficiently demonstrate the significance of that architecture. The improved efficiency and effectiveness appear marginal.
> >
> > [W3] The importance of value residual connection is proved. However, its novelty does not seem notable when compared to the shared attention map of ELAN.
> >
> > [W4] My concern has been fully addressed and resolved.
> >
> > [W5] I sincerely agree with the authors' claims as a researcher in this field. However, since only PSNR and SSIM are reported in this paper, it is inevitable that assessment for the performance gains must be based on these metrics. Moreover, compared to the improvements in efficiency, the performance gains are not significant enough for acceptance at this conference. That is, the time complexity should have been more reduced, or performance should have been more improved than other IR methods.
> >
> >
> > In summary, although the rebuttal addresses some of my concerns, the critical issues, including the urgency of the main problem, novelty, and performance gains, remain unresolved. Therefore, in my view, this paper still falls below the acceptance threshold.

---

> > > ### Author Response · Authors · 2023-11-23
> > >
> > > Perhaps I have misunderstood your valuable comments, you seem to think that as long as you use the modules used by previous generations it is not NOVEL enough, and at the same time you think that the model achieving a seven times acceleration ratio is equal or even slightly higher accuracy is still not enough to show the superiority of the method. I think that your views are too radical.
> > >
> > > 1. Regarding your accusation that the article is not novel enough. Our paper spends an entire section on the effect of residual connections on value, and shared attention is only one component of our work, not our main innovation. If you completely reject the value of the other modules of the paper because we used share attention, does that mean you think all papers using residual connections should be rejected because almost all networks use residual connections and they are not novel enough?
> > > 2. Regarding the point that a 7x speedup is still not enough to show the efficiency of a method when the accuracy is equal or slightly better. ShiftAddViT [1] achieved a 5.18x speedup on the GPU side with only equal accuracy through complex engineering optimizations, like TVM and binary operators, and the article was accepted by NeurIPS2023 without any problems. Maybe you misunderstood the time complexity reduction.
> > > 3. You repeatedly mention the importance of numerical metrics. I agree that numerical metrics are certainly one of the simplest means of comparing the merits of methods, but please note that our method is designed to be more efficient, not higher on PSNR or SSIM without seeing the qualitative results. If numerical metrics about performance were the only metrics used to evaluate whether the paper was accepted or not, should all articles that do not achieve a classification accuracy of 90 on the ImageNet1k dataset be rejected? Then, should all articles that do not achieve 66 on the IoU metric on the COCO dataset be rejected? Should all articles that do LLM be rejected because they do not perform as well as ChatGPT4?
> > > 4. Regarding your question about the reuse of attention maps between adjacent layers. One of the basic rules of nature is that everything is relative. In fact, reusing features in two consecutive layers can be an indirect way to convince other layers (e.g. layer 1 vs. layer 3) to be more separated in the optimization to ensure better performance. The presence of only feature redundancy in two consecutive layers is actually a sufficient condition for feature reuse in two consecutive layers, but it is not a necessary condition.
> > >
> > > In conclusion, we sincerely hope for a more comprehensive consideration of the potential contributions our work brings to the entire research community. We trust that you can provide a more rational and objective rating. Once again, we appreciate your time and efforts in reviewing our manuscript. Thank you!
> > >
> > > [1] You H, Shi H, Guo Y. ShiftAddViT: Mixture of Multiplication Primitives Towards Efficient Vision Transformer[J]. NeurIPS2023

---

### Author Response · Authors · 2023-11-19

We sincerely thank all the reviewers for their hard work in evaluating our work and providing valuable insights and suggestions. We have revised the main manuscript and supplementary materials to improve methodological clarity and coherence compared to previous versions. Please allow us to describe these changes in detail.

We have made the following changes to the main manuscript:

1. We made some changes to Figures 2 and 3. We have renamed the block composed of SPSA, GDFN and CSAU in the original paper as Share Attention Block (SAB) and provided a clearer and more complete explanation of how the network is constructed;
2. We have integrated Eq. 7 into Eq. 5, enabling reviewers and readers to better understand the entire SAB workflow;
3. We have further explained the specific mode of operation of CSAU and supplemented the ablation experimental results of this module.

We have made the following changes to the Appendices:

1. We use a table in Appendix B to further supplement the details of model training to help readers better reproduce the results of the paper;
2. We provide a visualization of the attention features of each layer in the SwinIR-L model in Appendix D, which helps readers to observe the redundancy of the attention features more clearly.

In addition, we have provided detailed responses to each reviewer's specific questions, addressing their concerns and incorporating recommendations where appropriate. We are committed to ensuring that our revised manuscripts reflect rigorous attention to reviewers' comments and to improving the quality and clarity of our work.

---

### Meta-Review · Area_Chair_mFFb · 2023-12-06

**Metareview:**

The paper under review has recieved mixed ratingss. Two of our reviewers, F9tc and nYLY, have evaluated the paper as slightly below the acceptance threshold, while another reviewer, DFQF, views it as marginally above the threshold. Notably, Reviewer DFQF, having a lower confidence level, did not offer substantial reasons to advocate for the paper's acceptance.

A significant part of the discussion revolved around the paper's novelty and the extent of its performance improvements in various tasks, as questioned by Reviewers F9tc and nYLY. These reviewers specifically pointed out concerns regarding several components of the approach, such as the application of residual on the value element, the sharing of attention maps, the CSAU component , and the group split strategy. It was noted that the concept of sharing attention maps has been previously introduced in ELAN. The authors countered by stating that shared attention is merely one aspect of their framework and not the main point of their contribution. However, the AC found that the shared attention block (SAB) was indeed highlighted as a key contribution in the introduction, raising questions about its overlap with ELAN, which was not adequately addressed in the main paper.

Regarding the residual on value element, while it is recognized as a novel addition, Reviewer F9tc observed that the improvements it offers are not uniformly evident across different tasks. The authors, in their rebuttal, demonstrated an improvement in PSNR by around 0.2dB, but this was only demonstrated in the 4x SR task on the Urban100 dataset.

The AC appreciates the additional experiments and ablation studies introduced during the rebuttal phase. However, these new insights necessitate another round of peer review to thoroughly examine the claims and details presented. The reviewers remain skeptical about the claims of novelty and the extent of performance enhancement. Considering the feedback and assessments of our reviewers, the decision has been made to reject the paper. For future improvement, the authors are advised to focus more on presenting the core contribution of the residual on V, offering a more comprehensive analysis of its mechanism and expanding the experimental evidence to demonstrate its effectiveness across a variety of tasks.

**Justification For Why Not Higher Score:**

Despite improvements and more results given during the discussion period, concerns about novelty and performance across tasks persist.

**Justification For Why Not Lower Score:**

NA

---

### Decision · Program_Chairs · 2024-01-16

Reject